# Intravital imaging-based genetic screen reveals the transcriptional network governing *Candida albicans* filamentation during mammalian infection

Rohan S Wakade[1], Laura C Ristow[1], Melanie Wellington[1], Damian J Krysan[1,2,3]*

[1]Department of Pediatrics, Carver College of Medicine, University of Iowa, Iowa City, United States; [2]Departments of Microbiology and Immunology, Carver College of Medicine, University of Iowa, Iowa City, United States; [3]Molecular Physiology and Biophysics, Carver College of Medicine, University of Iowa, Iowa City, United States

**\*For correspondence:**
damian-krysan@uiowa.edu

**Competing interest:** The authors declare that no competing interests exist.

**Abstract** *Candida albicans* is one of the most common human fungal pathogens. *C. albicans* pathogenesis is tightly linked to its ability to under a morphogenetic transition from typically budding yeast to filamentous forms of hyphae and pseudohyphae. Filamentous morphogenesis is the most intensively studied *C. albicans* virulence traits; however, nearly all of these studies have been based on in vitro induction of filamentation. Using an intravital imaging assay of filamentation during mammalian (mouse) infection, we have screened a library of transcription factor mutants to identify those that modulate both the initiation and maintenance of filamentation in vivo. We coupled this initial screen with genetic interaction analysis and in vivo transcription profiling to characterize the transcription factor network governing filamentation in infected mammalian tissue. Three core positive (Efg1, Brg1, and Rob1) and two core negative regulators (Nrg1 and Tup1) of filament initiation were identified. No previous systematic analysis of genes affecting the elongation step has been reported and we found that large set of transcription factors affect filament elongation in vivo including four (Hms1, Lys14, War1, Dal81) with no effect on in vitro elongation. We also show that the gene targets of initiation and elongation regulators are distinct. Genetic interaction analysis of the core positive and negative regulators revealed that the master regulator Efg1 primarily functions to mediate relief of Nrg1 repression and is dispensable for expression of hypha-associated genes in vitro and in vivo. Thus, our analysis not only provide the first characterization of the transcriptional network governing *C. albicans* filamentation in vivo but also revealed a fundamentally new mode of function for Efg1, one of the most widely studied *C. albicans* transcription factors.

## Editor's evaluation

Candida morphogenesis is important for virulence. This study provides important new information as to how *C. albicans* regulates the switch from budding to hyphal morphology. Their results identify transcription factors involved in the process of hyphal morphogenesis in the host. The results are convincing and will be interesting to scientists in the fields of medical mycology and cell biology.

## Introduction

Cellular morphology and morphogenesis are central features of fungal biology that contribute to the ability of fungi to cause disease in plants and mammals. In *Candida albicans*, one of the most

common human fungal pathogens, the ability to transition between round, budding yeast and filamentous hyphae and pseudohyphae is positively correlated with its ability to cause both mucosal and invasive infections (*Lopes and Lionakis, 2022*; *Sudbery, 2011*). Conversely, establishment of the commensal state of *C. albicans* within the gastrointestinal tract is negatively correlated with filamentation (*Witchley et al., 2019*). Furthermore, the yeast form is thought to be required for dissemination within the blood stream while transition to hyphae contributes to tissue damage in both disseminated and mucosal disease (*Saville et al., 2003*; *Meir et al., 2018*). Consequently, *C. albicans* morphogenesis has been an intensively studied virulence trait (*Villa et al., 2020*).

To date, the extensive literature devoted to understanding the mechanisms of *C. albicans* morphogenesis has been almost entirely focused on the in vitro induction of the yeast-to-hyphae transition (*Sudbery, 2011*; *Villa et al., 2020*). Based on these studies, a wide range of genes and regulatory pathways have been found to modulate this transition in response to various inducing conditions in vitro. A significant knowledge gap is that it is not clear how many of these genes and pathways are also involved in *C. albicans* filamentation in vivo. Our group has developed an in vivo imaging assay that allows us to directly, and quantitatively, characterize *C. albicans* filamentation during mammalian infection (*Wakade et al., 2021*; *Wakade et al., 2022b*). Specifically, a 1:1 mixture of WT and mutant *C. albicans* strains labeled with different colored fluorescent proteins are inoculated into subcutaneous/epithelial stroma of a mouse ear. Using confocal microscopy, the relative yeast/filament ratio and the filament length of the two strains is readily determined for large numbers of cells, providing a statistically powerful assay of in vivo filamentation. Importantly, we can characterize both initiation of the morphological switch (ratio of yeast to filament) and maintenance of filamentous growth (filament length). Furthermore, the tissue that is infected in this approach is anatomically equivalent to the subepithelial stroma below mucosal surfaces where *C. albicans* colonizes humans. During initial disease, *C. albicans* undergoes yeast-to-hyphae transition within this compartment and thus our approach models filamentation in a tissue that is directly relevant to *C. albicans* pathogenesis.

Our initial investigations using this model have revealed significant differences between the in vitro and in vivo regulation of *C. albicans* filamentation (*Wakade et al., 2021*; *Wakade et al., 2022a*). For example, the cAMP-protein kinase A pathway is required for initiation of filamentation in almost all in vitro conditions; specifically, strains lacking the adenylyl cyclase gene (*CYR1*) or the PKA catalytic subunits (*TPK1/2*) remain in yeast phase under hyphae-inducing conditions (*Cao et al., 2017*). In contrast, *cyr1ΔΔ* and *tpk1ΔΔ tpk2ΔΔ* mutants form hyphae in vivo to nearly the same extent as WT cells, indicating that this pathway plays a minor role in the regulating the switch between morphotypes in vivo (*Wakade et al., 2022a*). The length of the filaments formed by these mutants is shorter than WT (*Wakade et al., 2022b*). Consequently, the PKA pathway appears to switch from controlling initiation of hyphae formation in vitro to regulating maintenance of hyphal extension in vivo.

To further characterize the regulation of *C. albicans* filamentation in vivo, we screened a collection of 155 transcription factor (TF) deletion mutants (*Homann et al., 2009*) for those with either reduced hyphae/yeast ratios or filament length. We combined these studies with in vivo transcriptional profiling and genetic interaction analysis to provide the first model for the transcriptional network regulating *C. albicans* filamentation during infection of mammalian tissue. Although there are similarities between the TF networks operative in vitro and in vivo, we have identified TFs that affect filamentation in vivo but not in vitro. In addition, we provide evidence that the master regulator of *C. albicans* morphogenesis, Efg1, mainly functions to inhibit the function of Nrg1, a critical repressor of filamentation during filamentation within tissue.

## Results
### Identification of TFs required for *C. albicans* filament initiation in vivo

Previously, we reported the in vivo characterization of four TFs (Efg1, Brg1, Bcr1, and Ume6) with well-studied roles in vitro filamentation using both laboratory strains and clinical isolates (*Wakade et al., 2021*). As part of their characterization of a collection of 155 TF deletion mutants, Homann et al. reported that at least 40 mutants had effects, both positive and negative, on morphogenesis in vitro (*Homann et al., 2009*). Taking advantage of this important genetic resource, we set out to define the set of TFs that regulate *C. albicans* filamentation in vivo. To do so, we labeled all TF mutants by integrating an iRFP fusion protein at the *ENO1* locus as previously described (*Wakade et al., 2021*).

A 1:1 mixture of the iRFP-labeled TF mutant and SN250 (WT, unless otherwise indicated) containing a GFP fusion protein integrated at the same locus was injected into the pinna of a DBA/2 mouse ear. Twenty-four hours after inoculation, infected ears were imaged by confocal microscopy as detailed in the Methods and in a recent publication (*Wakade et al., 2022b*). This time point gives a ratio of yeast-to-filamentous cells similar to that resulting from incubation of the WT strain for 4 hr in RPMI medium supplemented with 10% fetal bovine serum (FBS); the latter conditions are those we used for in vivo to in vitro comparisons reported herein.

We characterized in vivo filamentation using two readouts that correspond to hyphal initiation and elongation (*Wakade et al., 2022b*; *Lu et al., 2014*). First, the ratio of yeast to filamentous cells was determined for each mutant; scoring criteria are outlined in Methods. This readout reflects the ability of a strain to initiate the filamentation program. Second, we estimated the filament length of the WT and TF deletion mutant cells. This readout reflects the ability of a strain to maintain filament elongation. We found that all TF mutants, with the exception of the *efg1ΔΔ* mutant, formed some proportion of filamentous cells. Each TF mutant was paired with the WT strain in the infection to better control for day-to-day variation. In addition, this experimental design improved the statistical power of the analyses because we did not need to correct for multiple comparisons. Previous control experiments have not found any evidence of in-trans effects associated with this co-infection approach (*Wakade et al., 2021*).

A volcano plot showing the normalized proportion of filamentous cells for each TF deletion mutant is shown in *Figure 1A*. A total of 19 TF mutants showed a statistically significant reduction (p<0.05, paired Student's t test) in the percentage of filaments relative the reference strain (*Supplementary file 1*). Only two mutants (*tup1ΔΔ* and *nrg1ΔΔ*) showed statistically significant increases in the proportion of filaments; Tup1 and Nrg1 are well-characterized repressors of filamentation in vitro (*Sudbery, 2011*). We classified the TF mutants with reduced filamentation into two groups: (1) core regulators of in vivo filamentation (*EFG1*, *BRG1*, *ROB1*), defined as mutants that formed fewer than 30% filamentous cells, and (2) auxiliary regulators, defined as mutants with a statistically significant change in filamentation (*Figure 1B*).

For the three core regulators and the two auxiliary regulators with the strongest phenotypes, we confirmed these ratios with independent experiments (*Figure 1C*). The *efg1ΔΔ* mutant formed essentially no filaments in vivo. Deletion of *BRG1*, *ROB1*, *TEC1*, and *RIM101* reduced the ratio of filaments/yeast but each of these homozygous deletion mutants formed filaments. Similar results were observed with these five mutants in vitro using RPMI+10% FBS medium at 37°C as inducing conditions (*Figure 1D*). Among the auxiliary regulators, a total of five TF deletion mutants were reported in the literature (*Homann et al., 2009*) to be deficient for filamentation in vitro (*TEC1*, *RIM101*, *CPH2*, *AHR1*, *ISW2*, *NDT80*, *SFL2*) while three mutants showed the opposite phenotype of increased filamentation in vitro but decreased filamentation in vivo (*FGR15*, *ZCF3*, *19.6874*). Finally, five TF deletion mutants had reduced filamentation in vivo but no reported effect on filamentation in vitro (*GRF10*, *SEF1*, *ZCF31*, *19.1150*, *19.2730*). The *stp2ΔΔ* mutant showed reduced filamentation as well; however, Stp2 regulates amino acid uptake (*Miramón and Lorenz, 2016*) and we suspected this phenotype may represent a genetic interaction with the *arg4ΔΔ* mutation present in this background (*Noble and Johnson, 2005*); consistent with this hypothesis, re-introduction of the *ARG4* gene to the *stp2ΔΔ* mutant restored normal filamentation (*Figure 1—figure supplement 1*).

Efg1 has been a well-studied master regulator of in vitro filamentation (*Glazier, 2022*). We had previously shown that deletion of *EFG1* in a set of four well-characterized clinical strains with varying levels of in vivo filamentation consistently eliminated filamentation (*Wakade et al., 2021*). These five strains were part of a larger collection of clinical isolates that were extensively characterized by *Hirakawa et al., 2015*. Among these strains, Hirakawa found one that contained a natural *EFG1* loss-of-function mutation (P94015). Since Efg1 is required for full virulence in mouse models and the ability to filament is strongly associated with virulence, we wondered if P94015 might contain a suppressor mutation that would allow it to filament in vivo. Consistent with its inability to filament in vitro and its lack of a functional copy of *EFG1*, the P94015 strain was essentially afilamentous in vivo (*Figure 1E*). These data indicate that Efg1 is an important regulator of filamentation within infected tissue.

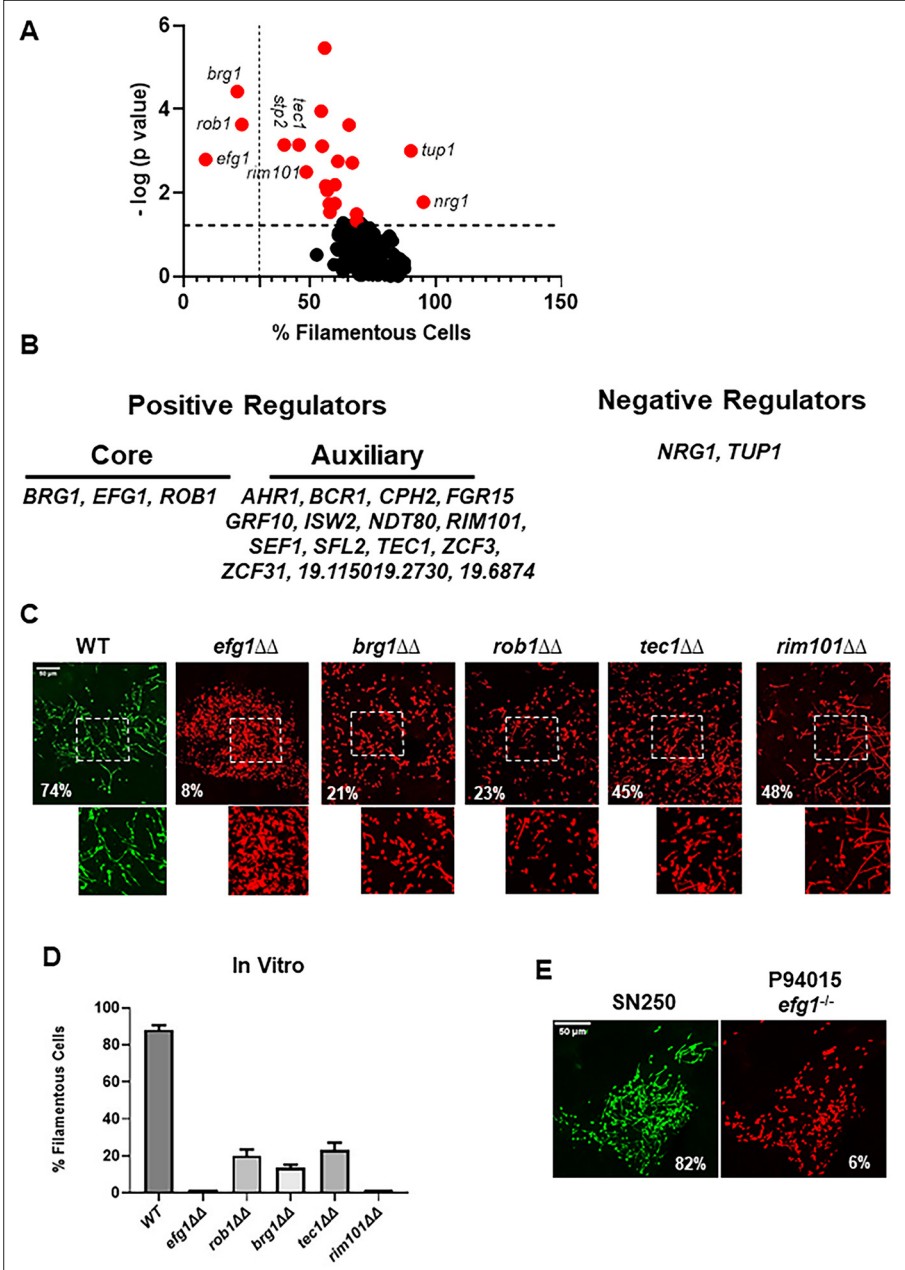

**Figure 1.** Intravital imaging assay identifies transcription factor mutants with altered initiation of filament formation in vivo. (**A**) Volcano plot of % filamentous cells of transcription factor homozygous deletion mutants normalized to co-infecting wild type (WT) cells 24 hr after injection into the subdermal tissue of a mouse ear. The p values are derived from paired Student's t tests of >100 cells in 3–4 fields per ear. See **Supplementary file 1** for all data. Red dots indicate mutants with statistically significant differences in % filamentous cells relative to WT. The vertical dotted line distinguishes the core positive regulators from auxiliary regulators; the horizontal dotted line indicates p=0.05 (Student's t test). (**B**) Gene names for the core and auxiliary positive regulators and the two negative regulators. (**C**) Representative images for WT and the five mutants with the strongest phenotypes; the percentage of filamentous cells is noted in each panel. The boxed regions shown are magnified in the lower panel. (**D**) The percentage of filamentous cells for the five mutants and WT after in vitro induction for 4 hr in RPMI+10% fetal bovine serum (FBS) at 37°C. The bars indicate mean of two independent experiments with >100 cells counted per strain. (**E**) Comparison of filamentation of the reference strain SN250 to a clinical isolate with a natural *EFG1* loss-of-function mutation (P94015).

The online version of this article includes the following source data and figure supplement(s) for figure 1:

*Figure 1 continued on next page*

*Figure 1 continued*

**Source data 1.** Source data for *Figure 1A*.

**Source data 2.** Confocal images for images shown in *Figure 1C* with file name as described in *Figure 1A*.

**Source data 3.** Bright-field images for the quantitative assessment of morphology shown in the graph in *Figure 1D*.

**Source data 4.** Confocal images for images shown in *Figure 1E* with file name as described in *Figure 1A*.

**Figure supplement 1.** The percentage of filamentous cells for *stp2ΔΔ arg4ΔΔ* and *stp2ΔΔ arg4ΔΔ::ARG4* indicates that *ARG4* and *STP2* genetically interact.

**Figure supplement 1—source data 1.** Images for the comparison of the *stp2ΔΔ* mutant with the ARG complemented *stp2ΔΔ* mutant.

## A distinct set of TFs regulates filament elongation in vivo

The filament lengths for all mutants, except the *efg1ΔΔ* mutant, were determined and a volcano plot for those data is shown in *Figure 2A*. A total of 59 TF deletion mutants had a statistically significant reduction in filament length (paired Student's t test, p<0.05) relative to WT strains (*Supplementary file 2*). Only two mutants, *nrg1ΔΔ* and *tup1ΔΔ*, showed increased filament length. Of the set of mutants with reduced filament length, 31 formed filaments that were 25% shorter than WT (*Figure 2B*); representative distributions of filament lengths for two of these mutants (*hms1ΔΔ* and *lys14ΔΔ*) are shown in *Figure 2C*. To our knowledge, there has been no systematic or large-scale genetic screen for mutants with reduced filament length. We, therefore, selected five TF deletion mutants (*HMS1, LYS14, SEF1, DAL81,* and *WAR1*) with strong defects in filament length but no known in vitro filamentation phenotypes. The five mutants formed filaments in vitro with lengths that were not significantly different than WT (*Figure 2D*).

Ume6 is a TF that is required for hyphal elongation in vitro (*Banerjee et al., 2008*) but its deletion mutant is not included in the Homann collection (*Homann et al., 2009*). We, therefore, constructed the *ume6ΔΔ* mutant in the SN background to determine if the reduced hyphal elongation phenotype was also observed in vivo. In vivo, the *ume6ΔΔ* mutant forms filaments to the same extent of WT but, consistent with its in vitro phenotype, the filaments are 40% shorter (*Figure 2E*). We had previously shown that deletion of *UME6* in clinical isolates has a minimal effect on filament initiation but we did not assess its role in elongation in that study (*Wakade et al., 2021*). As shown in *Figure 2—figure supplement 1*, re-analysis of those data revealed that deletion of *UME6* consistently leads to shorter filaments in the three clinical isolates that form filaments. This indicates that Ume6 functions mainly to regulate genes involved in hyphal elongation or maintenance rather than filament initiation in vivo.

As summarized in *Figure 2F*, our screen identified at total of 37 mutants that affected either initiation of filament formation (8 mutants), filament elongation (16 mutants), or both (15 mutants). Of the TFs that only affect initiation, 5 have in vitro phenotypes that match their in vivo phenotype; the 19.8874ΔΔ mutant shows increased filamentation in vitro while neither 19.2730 nor 19.1150 have been reported to affect filamentation. Of the TFs that only affect elongation, *HMS1, UME6,* and *ZCF29* (3/14) have been shown to have phenotypes in vitro. Thus, most of the elongation-associated TFs have not been found to have filamentation defects using standard in vitro assays. It is important to keep in mind that filament length has not been systematically studied prior to this work and, to our knowledge, no large-scale genetic characterization of genes affecting elongation has been reported. Lys14, Hms1, and Ume6, specific regulators of elongation in vivo, are required for virulence in disseminated candidiasis (*Banerjee et al., 2008*; *Pérez et al., 2013*), suggesting that this phenotype is relevant to pathogenesis, particularly since *LYS14* and *HMS1* mutants have few other in vitro phenotypes (*Homann et al., 2009*; *Shapiro et al., 2012*).

## Brg1, Efg1, and Rob1 have overlapping and distinct effects on the expression of filament-associated genes in vitro and in vivo

To further characterize the function of key in vivo regulators of the yeast-to-filament transition, we carried out focused in vitro and in vivo transcriptional profiling. In order to directly compare profiles in vivo and in vitro, we used the NanoString nCounter platform to query the expression of a set of 186 environmentally responsive genes previously used by the Mitchell and Filler labs to characterize the expression of *C. albicans* infecting mouse kidney tissue (*Xu et al., 2015*). The gene set contains

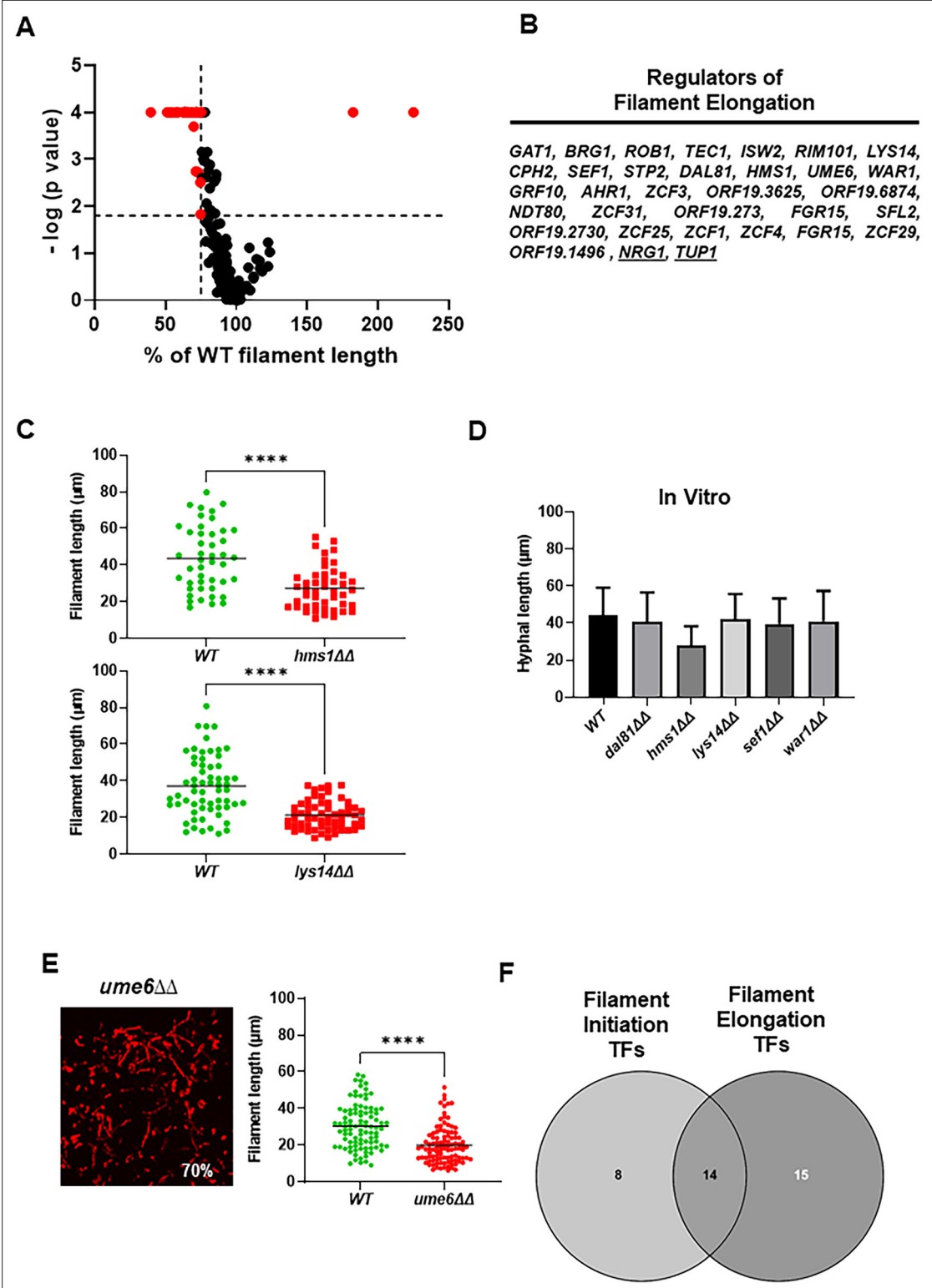

**Figure 2.** Identification of transcription factor (TF) deletion mutants with reduced filament length in vivo. (**A**) Volcano plot of filament length normalized to wild type (WT) in the same fields using the same images described in *Figure 1*. The p values are derived from a Mann-Whitney U test. The red dots indicate mutants with statistically significant reductions in filament length. The vertical dotted line indicates 75% of WT filament length; the horizontal dotted line indicates p=0.05 (Mann-Whitney). The full data set is provided in *Supplementary file 2*. (**B**) The set of TF mutants with lengths ≤75% of WT. (**C**) Representative data for two mutants with reduced filament length. **** indicates p value <0.001 by Mann-Whitney U test. (**D**) In vitro filament length for WT and five mutants with reduced filament length in vivo; 4 hr incubation in RPMI+10% fetal bovine serum (FBS) at 37°C. (**E**) The *ume6ΔΔ* mutant

*Figure 2 continued on next page*

*Figure 2 continued*

shows WT levels of filament formation but reduced filament length in vivo. (**F**) Venn diagram showing the distribution of mutants with reduced filament initiation, filament initiation and elongation, and elongation.

The online version of this article includes the following source data and figure supplement(s) for figure 2:

**Source data 1.** Confocal image files for transcription factor mutants with reduced filament length and its wild type (WT) comparator.

**Source data 2.** Prism files with the filament length measurements and statistical analysis for transcription factor mutants and their wild type (WT) comparators.

**Source data 3.** Images for the in vitro measurement of filament lengths shown in *Figure 2D*.

**Source data 4.** Image for length determination of *ume6ΔΔ*.

**Figure supplement 1.** The length of in vivo filaments formed by *ume6ΔΔ* mutants in four filament forming *C. albicans* strains.

**Figure supplement 1—source data 1.** Images for the measurement of *ume6ΔΔ* length in clinical strains.

57 hyphae-associated transcripts (30% of the total; see *Supplementary file 3* for complete list of genes and all raw and processed NanoString data). We recently reported the expression profile of WT infecting the ear and showed correlated with both the profile of cells from infected kidney tissue and in vitro hyphae induction; correlation between the two in vivo profiles was stronger than with in vivo (*Wakade et al., 2021*). These data indicate that the transcriptional and, by inference, the physiological state of the cells of *C. albicans* cells under these in vitro induction conditions is reasonably similar to the state of the cells in vivo. Consistent with that notion, the percentage of filamentous cells is very similar between 4 hr in vitro induction and 24 hr post infection (*Wakade et al., 2021*). Therefore, we feel that comparisons between these two conditions will be, to a first approximation, valid. Furthermore, these transcriptional similarities argue against our imaging model as being a special case that is not comparable to other infection niches or induction conditions.

We examined the effect of the three core regulators of filament initiation in vivo using NanoString, Consistent with their roles in both in vivo and in vitro filamentation, the *brg1ΔΔ*, *efg1ΔΔ*, and *rob1ΔΔ* mutants altered the expression of the environmentally responsive genes under both in vitro and in vivo conditions (*Figure 3A, B*, *Supplementary file 3*). From the chord plots, Efg1 is the dominant TF in vitro while Rob1 regulates a larger number of this set of genes in vivo. In contrast, Brg1 regulates a similarly sized set of genes under both conditions. Although there is overlap between the in vitro and in vivo sets of genes regulated by each TF, there are more genes that are regulated only in vitro or in vivo (*Figure 4A*). These data suggest that the direct and/or indirect target set of each TF is dependent on the environment. We also asked how many targets were regulated in common by the three core TFs. Focusing on the in vivo expression data, a significant percentage of genes downregulated in the *efg1ΔΔ* mutant (67%) was also downregulated in one of the other two mutants (*Figure 4B*). For the *rob1ΔΔ* mutant, the percentage of overlap was similar to the *efg1ΔΔ* mutant (72%) while all but five genes (83%) downregulated in the *brg1ΔΔ* mutant were downregulated in at least one of the other two mutants. A similar pattern of overlap is also evident for upregulated genes: *efg1ΔΔ* (62%); *rob1ΔΔ* (77%); and *brg1ΔΔ* (79%). Thus, the key regulators of filament initiation have overlapping effects on the expression of environmentally responsive genes in vivo and in vitro.

Brg1, Efg1, and Rob1 regulate the expression of other TFs required for the normal transition from yeast-to-filaments. In vitro, Efg1 affects the expression of six other regulators including both negative regulators (*Figure 4C*). Neither Rob1 nor Brg1 affect the expression of any other positive regulator of TF in our probe set but negatively regulate Nrg1. Only Efg1 affects the expression of *TUP1*. The regulatory relationships between Brg1, Efg1, and Rob1 remain the same in vivo (*Figure 4D*). Interestingly, Brg1 is the only factor that negative regulates Nrg1, although Efg1 plays a role indirectly through its activation of Brg1. In vivo, Brg1, Efg1, and Rob1 each positively regulate two other positive regulators, leading to a much more interconnected set of network interactions. Each of the less dominant TFs are regulated by two additional TFs. Efg1-Brg1-Tec1 and Efg1-Brg1-Ahr1 both appear to form feedforward loops. Thus, the network structure of the TFs affecting in vivo filament initiation is distinct from that present during in vitro filamentation.

## Efg1 and Brg1 interact with Tec1 to regulate initiation of filamentation

To further explore the functional consequences of these transcriptional interactions, we used a set of double mutants comprising all combinations of heterozygous deletion mutations in *EFG1*, *BRG1*,

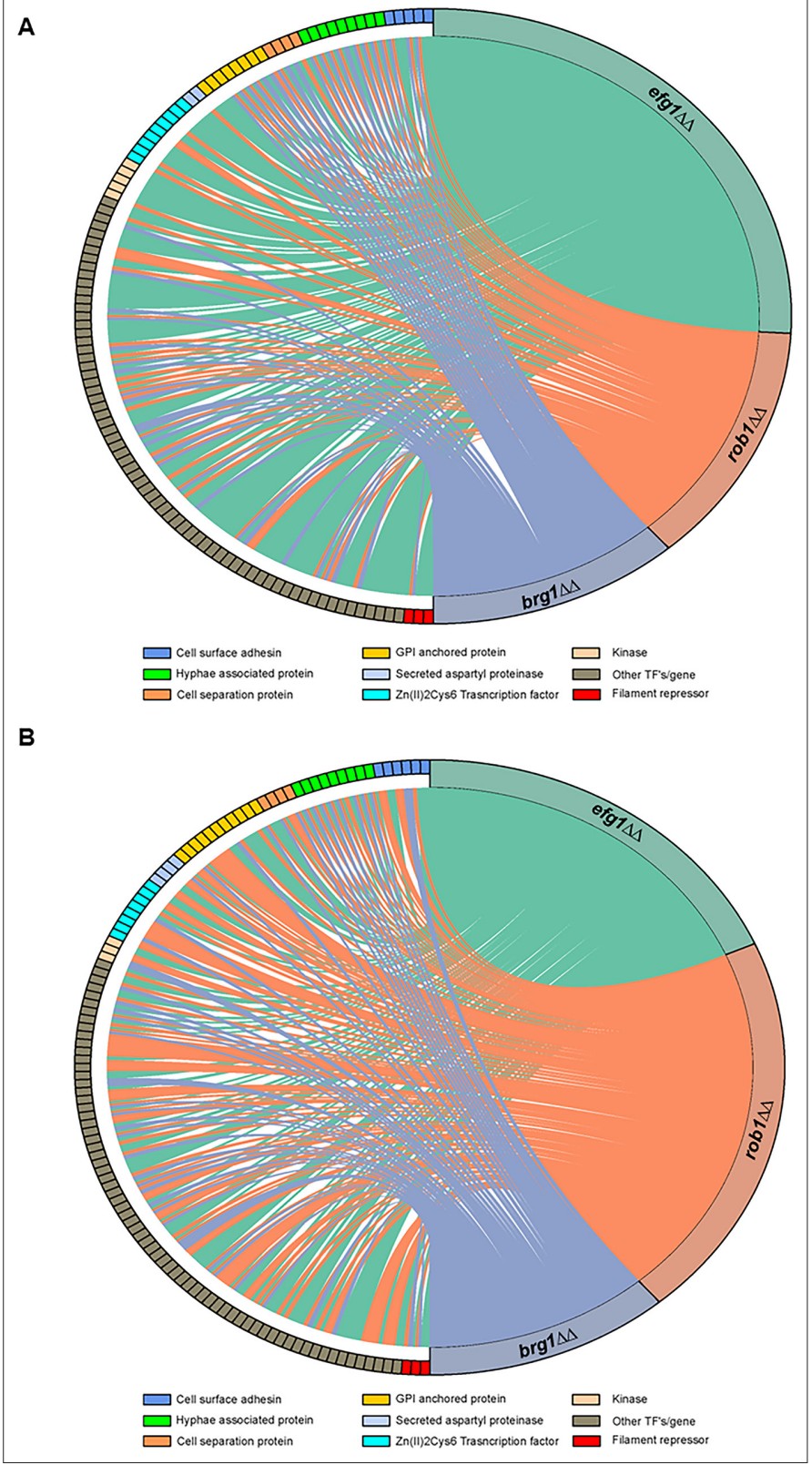

**Figure 3.** The core filament initiation regulators Brg1, Efg1, and Rob1 have distinct effects on the expression of environmentally responsive genes during filamentation in vitro and in vivo. Chord diagram depicts the sets of genes downregulated by twofold in the *brg1ΔΔ*, *efg1ΔΔ*, and *rob1ΔΔ* mutants relative to wild type (WT) during (**A**) in vitro (RPMI+10% fetal bovine serum (FBS), 37°C, 4 hr,) or (**B**) in vivo (24 hr post infection) filamentation.

*Figure 3 continued on next page*

*Figure 3 continued*

The expression of each target depicted on the left of the circle was determined using a NanoString nCounter; 188 genes were assayed of which 57 are known to be hyphae-associated. A line indicates that a given gene was reduced by twofold in a statistically significant manner (Benjamini-Yekutieli procedure, false discovery rate [FDR] = 0.1). The size of the colored regions on the right is proportional to the number of targets downregulated in the *efg1∆∆* (teal), *rob1∆∆* (salmon), and *brg1∆∆* (purple). The targets are arrayed on the left of the chord diagram and grouped by functional category as indicated by color code. The complete data sets are provided in *Supplementary file 3*.

*ROB1*, and *TEC1 Glazier et al., 2018*; for clarity, heterozygous mutants are indicated as having a single ∆ while homozygous mutants have a double ∆∆. We previously found that these four genes interact functionally during in vitro filament formation as measured by the proportion of filaments formed (*Figure 5A*, *Glazier et al., 2018*). In vivo, *EFG1* and *BRG1* interacted with *TEC1* (*Figure 5B, C*) while no other combinations showed a statistically significant change in the proportion of filaments formed. Thus, the network of functional interactions between these regulators in vivo is distinct from that observed in vitro. The functional genetic interactions between Efg1, Brg1, and Tec1 is consistent with the feedforward loop indicated by our in vivo transcriptional data (*Figure 4D*).

To further test the transcriptional overlap between Efg1 and Tec1, we generated in vivo transcriptional profiles for both the *efg1∆, tec1∆,* and *efg1∆ tec1∆* double mutants during ear infection. Although neither the *efg1∆* mutant nor the *tec1∆* mutant showed altered filamentation in vivo, both mutants affected gene expression with 43 (*efg1∆*) and 52 (*tec1∆*) genes differentially expressed (*Supplementary file 3*). As shown in the network diagram in *Figure 5D*, there is substantial overlap in the sets of genes regulated by the single mutants and the double mutant. The sets of genes downregulated in the single heterozygous mutants of *efg1∆* and *tec1∆* were very similar with 72% and 60% overlap, respectively (*Supplementary file 4*). Furthermore, the double heterozygous *efg1∆ tec1∆* mutant also showed significant overlap with the two single heterozygotes (83%), indicating that Tec1 and Efg1 regulate similar sets of genes.

The set of genes regulated by Efg1 and Tec1 contains TFs that affect in vivo filamentation as well as hypha-specific genes (*Supplementary file 4*). The functional interaction of Efg1 and Tec1 indicated that they are likely to regulate at least some of their targets synergistically or interdependently. To identify these targets, we used gene expression as a measure of the genetic interaction between Efg1 and Tec1. Using the multiplicative model for genetic interaction (*Figure 6A*), we calculated ε values for the interaction of *efg1∆ tec1∆* using the fold-change for a given target (*Glazier et al., 2018*; *Glazier and Krysan, 2020*). If Efg1 and Tec1 make independent contributions to the expression of a shared target, then ε=0; based on the range of standard deviations for the expression data, we estimated that ε = ±0.2 would not be distinguishable from 0. For negative interactions, ε≤–0.2 while positive interactions would have values ε ≥+0.2. A negative interaction indicates that Efg1 and Tec1 cooperatively regulate a given gene while no interaction indicates they regulate a gene independently (*Glazier and Krysan, 2020*). A positive interaction would suggest that Tec1 and Efg1 function in a linear pathway with a given gene. An example calculation for *TEC1* expression is shown in *Figure 6A* and indicates that *TEC1* expression is cooperatively dependent on both Efg1 and Tec1.

Ten of the TFs that affect filamentation in vivo were in our NanoString probe set. As shown in *Figure 6B*, Efg1 and Tec1 showed a negative genetic interaction with respect to the expression of 7 out of 10 TFs involved in in vivo filamentation. This negative interaction suggests that Efg1 and Tec1 are local hubs that interdependently regulate other filamentation-associated TFs in vivo. In contrast, Efg1 and Tec1 regulate hypha-specific genes independently because no ε values were outside of the ±0.2 range (*Figure 6C*). Taken together, these analyses support the regulatory circuit shown in *Figure 6D* in which: (1) Efg1 and Tec1 both regulate *TEC1* expression; (2) Efg1 and Tec1 cooperate to regulate the expression of other filamentation-associated TFs; and (3) Efg1 and Tec1 make independent contributions to the regulation of hypha-specific genes. Interestingly, the cooperative interactions between Efg1 and Tec1 are not observed in vitro (*Supplementary file 4*). Our data support the conclusion that the functional genetic interaction between Efg1 and Tec1 is due to their cooperative role in the regulation of other TFs that affect filamentation.

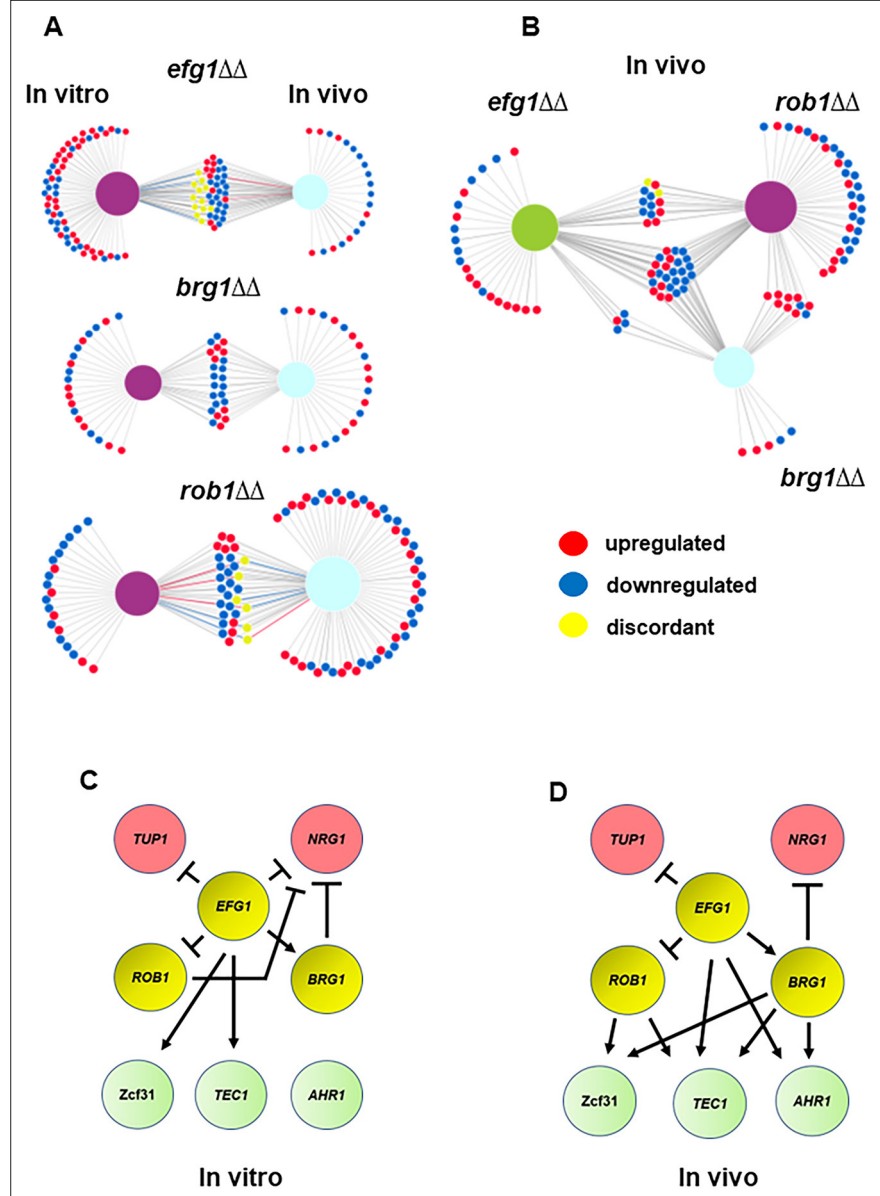

**Figure 4.** Core transcriptional regulators of in vivo filament initiation have overlapping and distinct regulons in vitro and in vivo. (**A**) Efg1, Brg1, and Rob1 regulate distinct sets of genes in vivo and in vitro. Differentially expressed genes were defined as those with ±2 fold-change relative wild type (WT) in the same condition with false discovery rate [FDR] = 0.1 as determined by the Benjamini-Yekutieli procedure. Raw data are available in *Supplementary file 3*. (**B**) Efg1 and Rob1 have distinct regulons in vivo while Brg1 regulated genes overlap extensively with those regulated by Rob1 and Efg1. (**C**) Regulatory networks of Efg1, Rob1, and Brg1 with negative and auxiliary regulators of filament initiation in vitro (**C**) and in vivo (**D**).

## TFs required for filament elongation in vivo regulate a common set of genes that does not include hyphae-associated transcripts

Of the TFs whose deletion mutants affected filament length, 15 have no effect on the yeast-to-filament transition. Ume6 regulates hyphal maintenance under in vitro conditions and has been considered one of the master regulators of hyphae formation. Consistent with the pioneering work of the Kadosh laboratory (*Banerjee et al., 2008*), loss of *UME6* has a significant effect on the length of filaments formed in vivo but no effect on the yeast-to-filament transition (*Figure 2E/F*). Because Ume6 regulates filament elongation under in vitro and in vivo conditions, we assessed the effect of *UME6* on gene expression in vivo. We also selected two additional TFs whose mutants have reduced filament

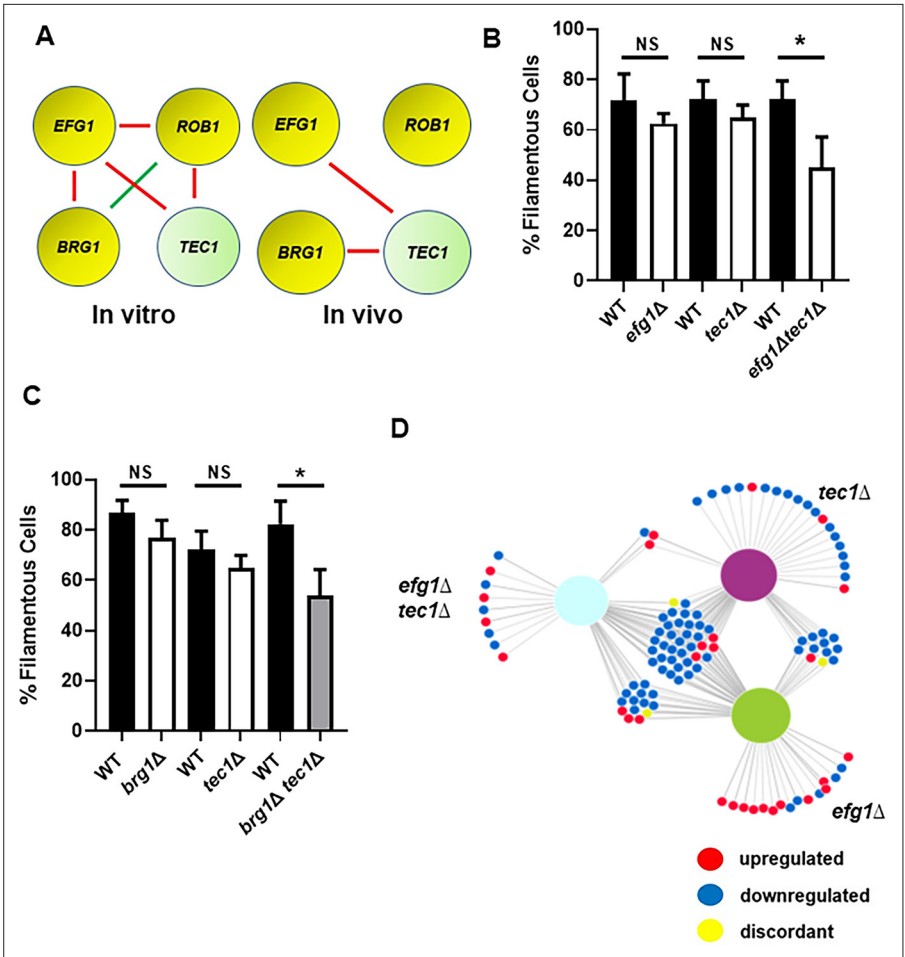

**Figure 5.** Efg1 and Brg1 show complex haploinsufficient genetic interactions with Tec1. (**A**) The genetic interactions of the core filament initiation regulators with Tec1 differ between in vitro and in vivo conditions. Red indicates a negative interaction; green indicates a positive interaction according to the multiplicative model described in text. The single Δ after the gene name indicates that the mutant is heterozygous at that locus. Filament initiation data for the interaction of (**B**) *efg1Δ* and *tec1Δ* and (**C**) *brg1Δ* and *tec1Δ*. *p<0.05 paired Student's t test. Bars indicate mean of 4–5 replicate fields with >100 cells. Error bars indicate standard deviation. (**D**) The differentially expressed genes in the *efg1Δ tec1Δ* double heterozygous mutant overlap with the genes differentially expressed in the corresponding single heterozygous mutants (*efg1Δ* and *tec1Δ*). Full data set is provided in **Supplementary file 4**.

length but no change in filament initiation to study: *HMS1*, a TF previously found to affect filamentation at high temperature (42°C, **Shapiro et al., 2012**) in vitro, and *LYS14*, a TF with no morphology or other in vitro phenotype (**Homann et al., 2009**; **Pérez et al., 2013**). Although *LYS14* is homologous to *Saccharomyces cerevisiae* TFs involved in the regulation of lysine biosynthesis genes, this function is not conserved in *C. albicans* and little else is known about its role in *C. albicans* biology (**Pérez et al., 2013**). Deletion mutants of *HMS1* and *LYS14* have reduced competitive fitness in a mouse model of disseminated candidiasis while the *hms1ΔΔ* mutant is also less fit during GI colonization (**Pérez et al., 2013**). Strains lacking *UME6* have reduced virulence in the disseminated infection model but are hyper-fit during GI colonization (**Witchley et al., 2019**).

Neither Ume6, Lys14, nor Hms1 have a consistent effect on the expression of most hyphae-associated genes in vivo (**Figure 7A**, **Supplementary file 5**), particularly when compared with the core regulators of filament initiation. Of the hyphae-associated genes, *HYR1* and *HWP1* are the only gene downregulated by >2-fold in the deletion mutants of all three elongation-associated TFs. Of the three TFs involved in hyphal elongation, Ume6 plays the most prominent role in the expression of hyphae-associated genes (**Figure 7A**) and affects the expression of 6/9 genes. *UME6*, *LYS14*, and

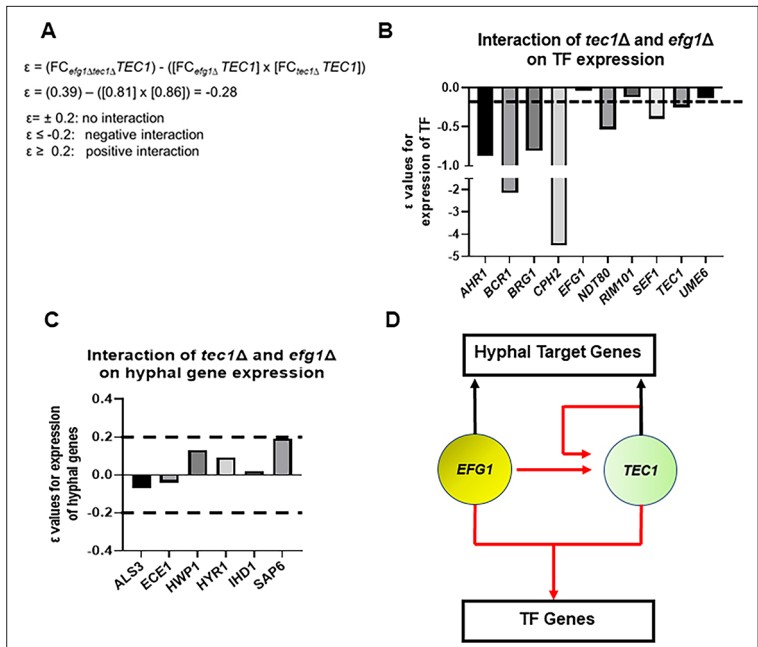

**Figure 6.** Efg1 and Tec1 synergistically affect the expression of auxiliary regulators of filament initiation but independently regulate the expression of hypha-associated genes. (**A**) Sample calculation of the interaction score (ε) for the effect of the *tec1Δ*, *efg1Δ*, and *tec1Δ efg1Δ* mutants on auxiliary transcription factor (TF) expression and hyphae-specific genes. ε values calculated using data in ***Supplementary file 4***. The dotted line indicates the cut-off for negative or positive interactions (ε±0.2) based on the estimated error; targets for which the –0.2>ε<0.2 were considered to have no interaction. (**B**) depicts interactions for auxiliary TFs and (**C**) for a set of hyphae-associated genes. (**D**) Proposed regulatory circuit for Efg1 and Tec1 based on genetic interaction analysis.

*HMS1* mutants do not affect the expression of the core TFs involved in filament initiation. We further analyzed the overlap between genes regulated by Rob1, the filament initiation-associated TF that regulates the largest set of genes in vivo, with the filament elongation TFs. The *hms1ΔΔ* and *lys14ΔΔ* mutants have minimal overlap with the genes regulated by Rob1, while nearly half the genes regulated by Ume6 are shared with Rob1 (***Figure 7B***), although that represents less than 25% of the genes regulated by Rob1. In vivo, Ume6, Hms1, and Lys14 are required for the full expression of 32, 23, and 29 genes, respectively. Hms1 and Lys14 have 17 genes in common (***Figure 7C***); Lys14 has only 5 genes which it uniquely regulates. Ume6 has the largest set of uniquely regulated genes (16 genes). Although Ume6, Lys14, and Hms1 all selectively affect elongation, Ume6 does seem to share more targets with hyphae initiation-associated TFs relative to Lys14 and Hms1.

The NanoString probe set was not specifically designed to include genes that are important for hyphal elongation. However, the set of genes regulated by two or more of the elongation regulators provides some insight into transcripts important during this phase of filamentation. Specifically, 9/23 genes (*ALS2, ALS9, CRH11, RBT1, SAP1, SAP10, SCW4, SCW11, SOG2*; ***Supplementary file 5***) regulated by at least two of hyphal elongation TFs are cell wall or secreted proteins. This correlation suggests that elongation-associated TFs regulate cell wall remodeling during the elongation phase of filamentation. Overall, our profiling of these mutants indicates that they regulate a distinct set of genes relative to TFs required for the initiation of filamentation.

## Deletion of *NRG1* but not *TUP1* suppresses the filamentation defect of *efg1ΔΔ* in vivo

*C. albicans* filamentation requires the ordered activity of both positive and negative regulators of gene expression (***Sudbery, 2011***; ***Lu et al., 2014***). In vitro, repressors of *C. albicans* filamentation include Nrg1, Tup1, Ssn6, and Rfg1. Although Homann et al. found that 16 TF deletion mutants were hyper-filamentous on at least one condition in vitro (***Homann et al., 2009***), our in vivo screen of the same deletion set found that the *nrg1ΔΔ* and *tup1ΔΔ* mutants were the only two with statistically

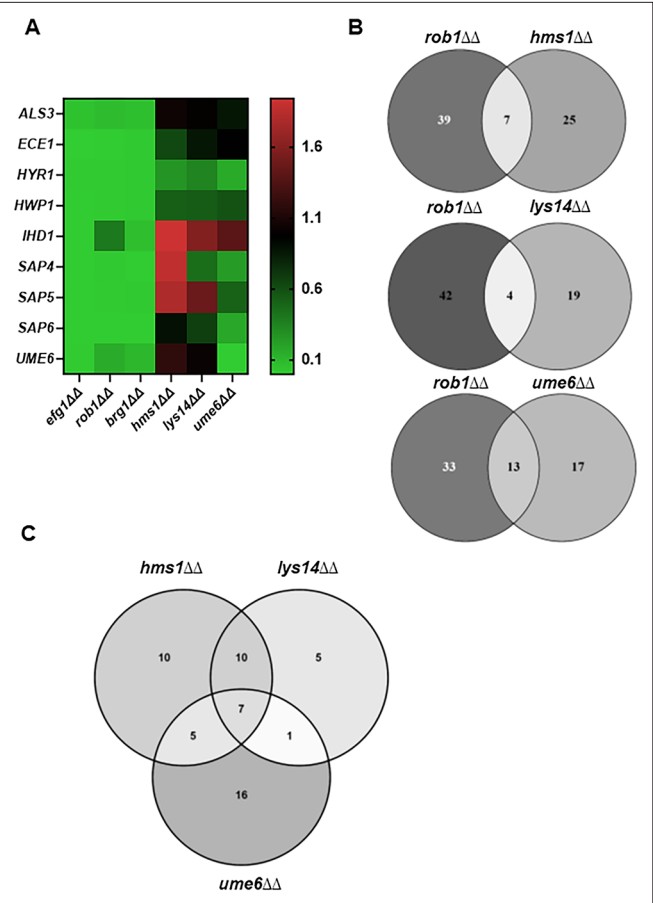

**Figure 7.** Transcription factors that specifically affect filament length regulate a set of genes distinct from those regulated by transcription factors involved in filament initiation. (**A**) Heat map comparing the expression of the indicate hypha-associated genes in the core filament initiation regulators (*efg1ΔΔ*, *brg1ΔΔ*, and *rob1ΔΔ*) with three regulators of hyphal elongation (*hms1ΔΔ*, *lys14ΔΔ*, and *ume6ΔΔ*). The full data set is provided in **Supplementary file 5**. (**B**) Venn diagrams comparing genes downregulated in the *rob1ΔΔ* mutant with three regulators of hyphal elongation (*hms1ΔΔ*, *lys14ΔΔ*, and *ume6ΔΔ*). (**C**) Venn diagram comparing the overlap in genes downregulated in *hms1ΔΔ*, *lys14ΔΔ*, and *ume6ΔΔ*.

significant increases in percentage filamentous cells. Similarly, deletion mutants of *NRG1* and *TUP1* were the only two that formed longer filaments than the WT strain. We were interested, therefore, in generating double mutants of three key positive regulators (Brg1, Efg1, and Rob1) and repressors (Tup1 and Nrg1) to gain information about how they worked together to regulate morphogenesis in vivo.

We first examined the genetic interaction between the Tup1 and Efg1. The *tup1ΔΔ* mutant forms nearly 100% filaments in vitro and in vivo while the *efg1ΔΔ* mutant forms almost no filaments under either condition. The *tup1ΔΔ efg1ΔΔ* double mutant forms approximately 20% filamentous cells in vitro and in vivo (**Figure 8A**), indicating the relief of Tup1 repression requires Efg1 for filamentation to proceed. Next, we constructed homozygous deletion mutants of *BRG1*, *EFG1*, and *ROB1* in the *nrg1ΔΔ* background (**Figure 8B**). In vitro, the *nrg1ΔΔ* mutant forms predominantly hyphae while *nrg1ΔΔ* double mutants with the three positive regulators formed very few hyphae with pseudohyphae as the dominant morphotype (**Figure 8C**). In vivo, all three double mutants formed similar numbers of filaments as wild type (WT) cells (**Figure 8D, E**) whereas the single mutants of *BRG1*, *EFG1*, and *ROB1* are significantly deficient (**Figure 1C**). Thus, deletion of *NRG1* suppresses the in vivo filamentation defect of all three of the major TF regulators of in vivo filamentation.

We noticed that the *nrg1ΔΔ* mutants showed increased branching relative to WT cells in vivo. Increased branching of filaments is a characteristic of pseudohyphae (**Sudbery et al., 2004**). In our

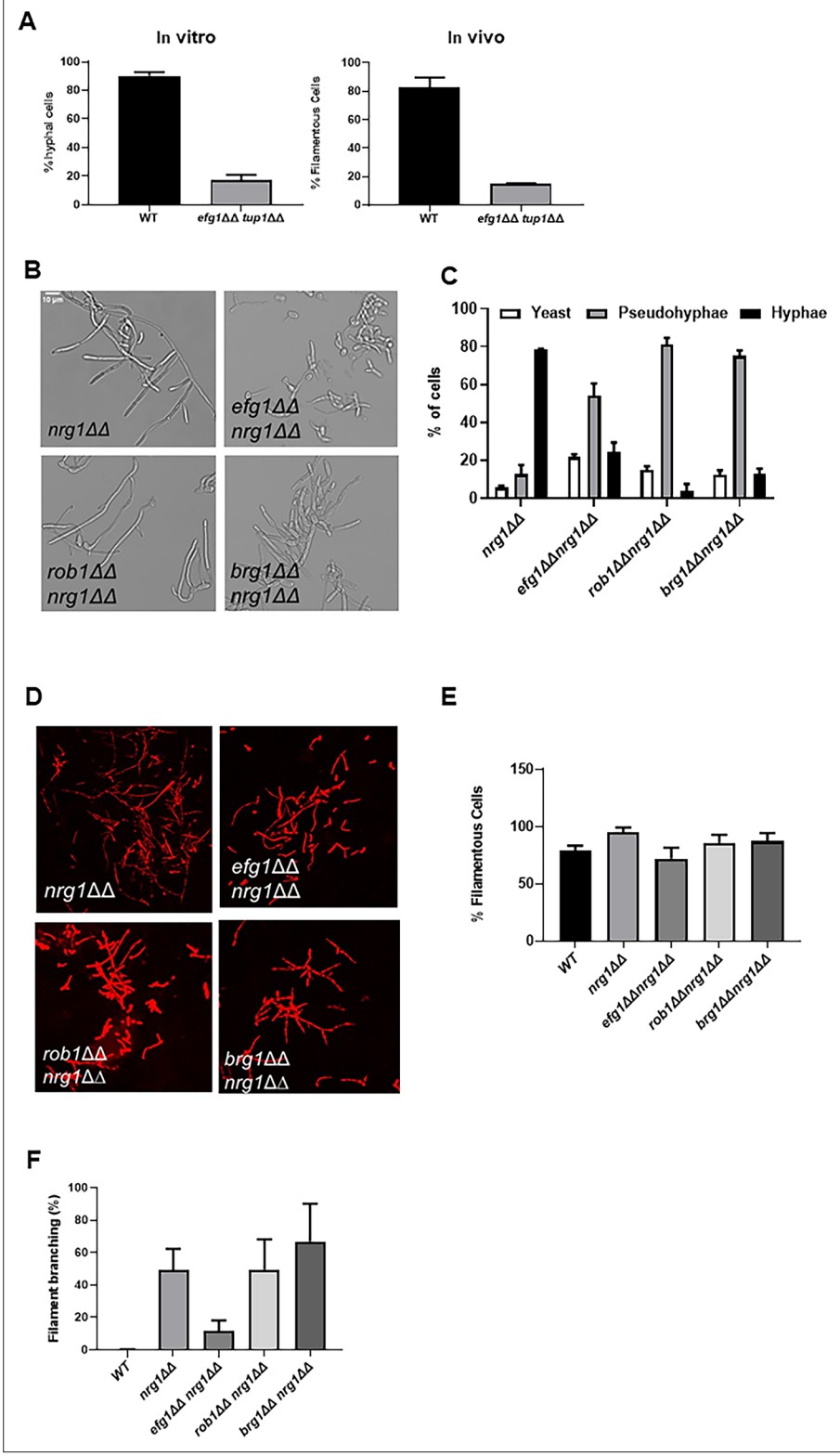

**Figure 8.** Genetic interactions between the core positive and negative regulators of filament initiation. (**A**) The percentage of filaments formed by the *tup1ΔΔ efg1ΔΔ* mutant relative to wild type (WT) after 4 hr induction with RPMI+10% fetal bovine serum (FBS) or 24 hr infection in mouse ear. (**B**) Photomicrographs of the indicated mutants after in vitro induction. (**C**) Distribution of yeast, hyphae, and pseudohyphae after in vitro induction. (D) Representative micrographs of in vivo filamentation for the indicated strains. (**E**) The percentage of filaments

*Figure 8 continued on next page*

*Figure 8 continued*

observed in vivo for the indicated strains. (**F**) The percentage of cells with branches observed in vivo. The bars indicate at least two independent inductions and ear assays with standard deviation indicated by error bars.

The online version of this article includes the following source data for figure 8:

**Source data 1.** *Figure 8A and B*.

**Source data 2.** Confocal images corresponding to the in vivo in *Figure 8D*.

experience, WT cells branch very little in vivo, suggesting that hyphae are the predominant morpho-type at the time point that we are analyzing (*Figure 8F*). Deletion of *NRG1* increases the extent of branching to almost 50% (*Figure 8F*). The branching phenotype of *nrg1ΔΔ* is preserved in the *brg1ΔΔ nrg1ΔΔ* and *rob1ΔΔ nrg1ΔΔ* mutants, while the *efg1ΔΔ nrg1ΔΔ* mutant modest branching relative to *nrg1ΔΔ*. This indicates that the filaments formed by the *efg1ΔΔ nrg1ΔΔ* mutant are more hyphae-like than the *nrg1ΔΔ* mutant.

A possible explanation for the reduced branching of the *efg1ΔΔ nrg1ΔΔ* mutant relative to the *nrg1ΔΔ* single mutant is that the TF *ACE2* is expressed at very high levels in the double mutant (FC 13.8 relative to WT; q value = 0.002, *Supplementary file 6*). We have previously shown that Efg1 suppresses *ACE2* expression in vitro and binds to the promoter of *ACE2* (*Saputo et al., 2014*); *ACE2* expression is also increased in the *efg1ΔΔ* mutant in vivo (FC 6.1 relative to WT q value = 0.04, *Supplementary file 3*). We have also demonstrated that Ace2 suppresses lateral yeast formation during in vitro filamentation (*Wakade and Krysan, 2021*). Thus, the increased expression of *ACE2* in the *efg1ΔΔ nrg1ΔΔ* mutant suggests that loss of Efg1 leads to increased *ACE2* expression which in turn reduces lateral branching.

## Efg1, Brg1, and Rob1 play distinct roles in the regulation of hyphae-associated gene expression after relief of Nrg1 repression

After relief of Nrg1 repression, it is not currently clear which TFs positively regulate the transcriptional program repressed by Nrg1. To explore this question, we compared the expression profiles of the *efg1ΔΔ nrg1ΔΔ*, *brg1ΔΔ nrg1ΔΔ*, and *rob1ΔΔ nrg1ΔΔ* double mutants to single mutants of each positive regulator of filamentation (*Supplementary file 6*). We focused on the set of nine hypha-associated genes listed in *Figure 7A* and the data for *ALS3*, *ECE1*, and *HWP1* from both in vitro and in vivo conditions are shown in *Figure 9A* while data for the remaining genes are shown in *Figure 9—figure supplement 1*. The expression of *ALS3*, *ECE1*, and *HWP1* are all significantly reduced in the TF single mutants. In the case of the *efg1ΔΔ nrg1ΔΔ* mutant, the expression of all three canonical hypha-associated genes is restored to WT levels; of the remaining genes, only *SAP6* expression is reduced in the *efg1ΔΔ nrg1ΔΔ* mutant (*Figure 9—figure supplement 1*). This is a rather surprising result because three separate groups have found that Efg1 directly binds the promoters of these genes (*Lassak et al., 2011*; *Witchley et al., 2021*; *Do et al., 2022*). However, our data indicate that Efg1 is not necessary for the expression of *ALS3*, *ECE1*, and *HWP1* as well as other hypha-specific genes after relief of Nrg1 expression. Rob1 and Brg1, on the other hand, play a consistent role in the expression of the hypha-associated genes in vivo and in vitro with Rob1 having a more important role for most of the genes that we analyzed (*Figure 9A* and *Figure 9—figure supplement 1*). In terms of regulation of hypha-induced genes, our data suggest that Efg1 functions upstream of Nrg1 and may mediate the relief of Nrg1 repression rather than the direct induction of hypha-specific gene expression after relief of Nrg1 repression.

Although Efg1 does not appear to contribute directly to the expression of canonical hyphae-associated genes after relief of Nrg1 repression, 49 genes have reduced expression in the *efg1ΔΔ nrg1ΔΔ* mutant relative to WT in vivo as indicated by the Venn diagram in *Figure 9B*. Of these 49 genes, the promoters of nearly 50% (22/49) have been shown to be bound by Efg1 in vitro while only three have Nrg1 Responsive Elements (NRE, *Murad et al., 2001*) in their promoter regions (*Figure 9B*). In contrast, all Efg1 bound genes whose expression was restored to WT levels in the *efg1ΔΔ nrg1ΔΔ* mutant deletion contained NRE sequences in their promoter (*Figure 9B*), suggesting that they are directly bound by Nrg1. From these data, Efg1 and Nrg1 appears to bind a common set of hypha-associated genes during filament initiation but Efg1 is not required for their expression. Most likely, Rob1 and Brg1 along with additional TFs contribute to the expression of Nrg1-repressed,

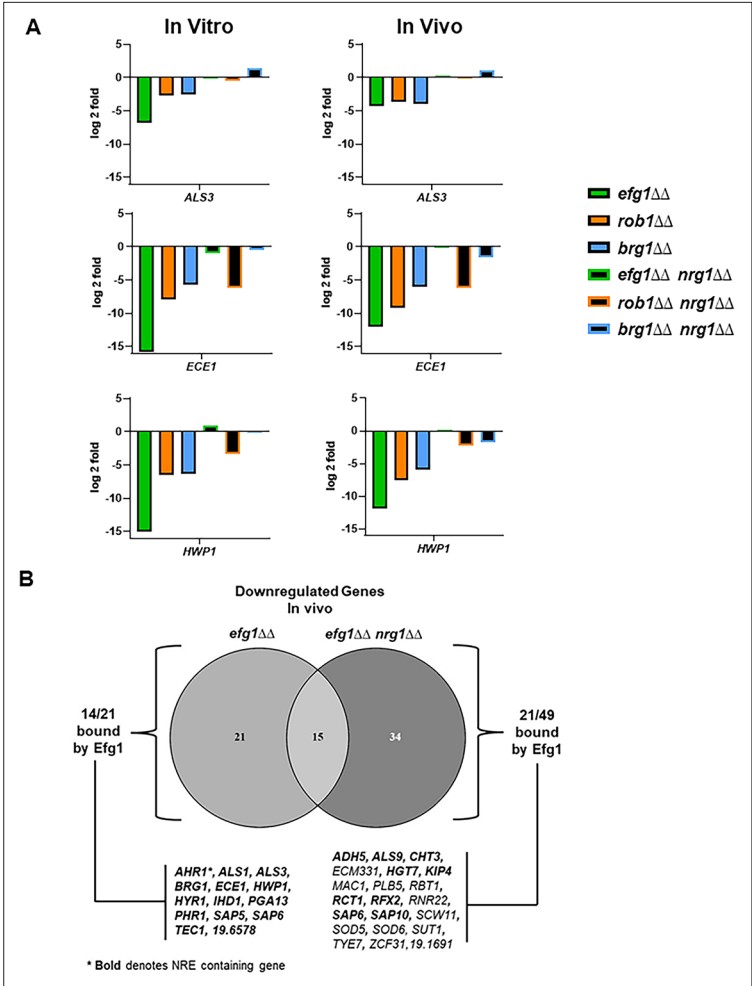

**Figure 9.** Deletion of *NRG1* restores the expression of hypha-associated genes to the *efg1ΔΔ* mutant in vitro and in vivo. (**A**) The expression of *ALS3*, *ECE1,* and *HWP1*, canonical hypha-associated genes, in the indicated strains. The fold-change is relative to wild type (WT) for both in vitro and in vivo inductions The data are presented in *Supplementary file 3* and *Supplementary file 6* along with standard deviation and false discovery rate (FDR) statistics. (**B**) The Venn diagram compares the downregulated genes in the *efg1ΔΔ* and *efg1ΔΔ nrg1ΔΔ* mutants (see *Supplementary files 3; 6 and 7*). Direct Efg1 targets are genes identified in references (*Lassak et al., 2011*; *Witchley et al., 2021* or *Do et al., 2022*). Nrg1 Response Elements (NRE) were as defined in reference (*Murad et al., 2001*).

The online version of this article includes the following figure supplement(s) for figure 9:

**Figure supplement 1.** The expression of five canonical hypha-associated genes in the indicated strains.

hypha-associated genes. After relief of Nrg1 repression, Efg1 binds to the promoters and regulates the expression of a second set of genes that are not repressed by Nrg1. Since canonical hyphae-specific genes are in the first class of Efg1 targets, our data are consistent with the surprising result that Efg1 indirectly regulates hyphae-specific genes, possibly by mediating the displacement of Nrg1 from their promoters.

## Discussion

*C. albicans* filamentation is one of the most extensively studied virulence traits of this important human fungal pathogen (*Sudbery, 2011*; *Villa et al., 2020*; *Homann et al., 2009*; *Hirakawa et al., 2015*). Prior to this work, systematic large-scale genetic analysis of filamentation was limited to in vitro induction of filamentation and very few in vivo studies had been reported (*Homann et al., 2009*; *Uhl et al., 2003*; *Noble et al., 2010*; *O'Meara et al., 2015*). The in vivo analysis of *C. albicans*

filamentation using the direct inoculation of the subepithelial/subdermal tissue of mouse pinna followed by confocal microscopy reported herein has several features (*Wakade et al., 2021*; *Wakade et al., 2022b*; *Wakade et al., 2022a*). First, the method uses direct inoculation of the tissue and thus is not confounded by ability of mutants to colonize, disseminate, and/or invade target organs. Second, the subepithelial tissue is anatomically similar to the stroma below mucosal surfaces and is relevant to infection. Third, hypha-associated gene expression in infected ear tissue correlates well-infected kidney tissue (*Wakade et al., 2022a*), the main target of disseminated *C. albicans* infections in mice. Fourth, as we have demonstrated, the model allows the characterization of both filament initiation and elongation.

Efg1, Brg1, and Rob1 have the strongest phenotypes with respect to both filament initiation and hyphal elongation, although the *efg1ΔΔ* mutant does not form filaments that can be assessed. These three TFs have similar phenotypes in vivo and in vitro (*Figure 1*); additionally, *efg1ΔΔ* and *brg1ΔΔ* mutants have reduced in vitro and in vivo filamentation across multiple clinical isolates (*Wakade et al., 2021*). The filamentation phenotypes of these three mutants have also been studied during GI colonization using a FISH-based assay (*Witchley et al., 2019*). During GI colonization, the *efg1ΔΔ*, *brg1ΔΔ*, and *rob1ΔΔ* mutants were predominantly in yeast form suggesting they are important for filamentation in both niches. In contrast, the *tec1ΔΔ* mutant formed near WT proportions of filaments during GI colonization (*Witchley et al., 2019*) while the proportion of filaments were reduced in the ear infection model, indicating that Tec1 may play a more important role in filamentation in the submucosae compared to colonization. Finally, Witchley et al. observed that the *ume6ΔΔ* mutant formed filaments to an extent similar to WT in the gut (*Witchley et al., 2019*). We also observed no difference in the proportion of filaments formed by the *ume6ΔΔ* mutant relative to WT but that the filaments were much shorter. Witchley et al. did not measure filament length in the GI model but inspection of their micrographs suggests that *ume6ΔΔ* filaments were comparable in length to WT (*Witchley et al., 2019*). Thus, Efg1, Brg1, and Rob1 are important regulators of filamentation in these two niches whereas the roles of other TFs appear to vary from niche to niche.

In vitro analysis of filamentation/hyphae formation has been performed using a wide range of induction conditions and specific genes are required for filamentation in specific conditions. For example, Homann et al. screened the TF library that we used here using solid agar plates and 15 different media (*Homann et al., 2009*). Deletion mutants of *BRG1*, *EFG1*, *ROB1* showed reduced filamentation on the majority of media/conditions while deletion of the repressors *TUP1* and *NRG1* showed increased filamentation under all condition. Our in vivo results for filament initiation correlate well with these trends indicating that TFs with important roles in vitro also have important roles in vivo. The mutants with less severe phenotypes in vivo such as *tec1ΔΔ*, *ndt80ΔΔ*, and *cph2ΔΔ*, for example, also had phenotypes in vitro but these were not as consistent across the different conditions. We suggest that these auxiliary regulators respond to a more limited set of stimuli or function together with one or more additional TFs. The synergistic interactions of Efg1 and Brg1 with Tec1 is an example where two core TFs function with an auxiliary TF to, in the case of Efg1-Tec1, interdependently regulate other auxiliary regulators (*Figure 6D*). Under some in vitro conditions, TFs such as Tec1 and Cph2 have very strong filamentation phenotypes (*Homann et al., 2009*; *Lane et al., 2001*) and we suggest that this may occur because those in vitro conditions generate a very specific filamentation signal that is highly dependent on a specific TF(s). In vivo, on the other hand, this signal is but one of many and therefore has a modest effect on filamentation process.

In the case of repressors of filamentation, a similar pattern arises where Tup1 and Nrg1 repress filamentation under nearly all in vitro conditions; in vivo, *tup1ΔΔ* and *nrg1ΔΔ* mutants were the only two with increased filamentation relative to WT. In vitro, 19 TF mutants have increased filamentation and, consequently, the largest discrepancy between in vitro and in vivo regulators is within the set of repressors (*Homann et al., 2009*). Of these, Nrg1, Tup1, and Ssn6 have been characterized as bona fide transcriptional repressors (*Murad et al., 2001*; *Braun and Johnson, 1997*; *García-Sánchez et al., 2005*) while the other TF mutants with increased filamentation are more likely to indirectly cause increased filamentation by altering the physiology of the cell such that its filamentation is a compensatory response. Although our assay may have reduced sensitivity for repressors, it is also possible that the in vivo environment is such a strong inducer of filamentation that mild to moderate repressor phenotypes are masked in this setting.

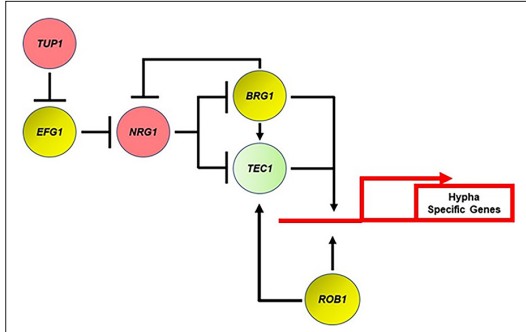

**Figure 10.** Transcriptional regulatory circuit for the expression of hypha-specific genes. Red indicates repressor of filamentation; yellow indicates core positive regulator; and light green indicates auxiliary. Arrows indicate positive regulatory step and perpendicular line indicates negative regulatory step.

Nrg1 and Tup1 are widely described as functioning together to repress filamentation (*García-Sánchez et al., 2005*). Tup1 lacks a DNA-binding domain while Nrg1 does bind specific NRE-binding sites. However, while transcriptional profiling of *tup1ΔΔ* and *nrg1ΔΔ* mutants during filamentation showed significant overlap between the two regulons (*García-Sánchez et al., 2005*), distinctions in the differentially expressed genes were apparent as well. Furthermore, constitutive expression of *NRG1* in the TET-off system blocks filamentation under in vitro inducing conditions (*Saville et al., 2003*), while expression of *TUP1* from the same promoter does not (*Ruben et al., 2020*). These observations indicate that Tup1 and Nrg1 have genetically separable functions. Our data further support and extend this conclusion. The increased filamentation induced by deletion of *TUP1* is dependent upon Efg1 (*Figure 8A*),

under in vitro and in vivo conditions. In contrast, filamentation of the *nrg1ΔΔ* mutant is not dependent on Efg1 (*Figure 8D*). The differential Efg1 dependence of Tup1 and Nrg1 is hard to reconcile with a model in which these two repressors function as a complex. Ssn6 and Tup1 are also a well-characterized co-repressor pair that is conserved in eukaryotic cells and have been proposed to function with Nrg1 as complex (*García-Sánchez et al., 2005*). However, we observed no phenotype for the *ssn6ΔΔ* mutant in vivo; in vitro, the increased filamentation for *ssn6ΔΔ* is not consistently observed. Taken together, our data and those in the literature suggest that Tup1 and Nrg1 play distinct roles in the repression of filamentation and that the mechanisms mediating relief of their repression are also distinct.

A model for the interactions of Tup1, Efg1, and Nrg1 that is consistent with our data as well with literature data is as follows. Tup1 lacks a DNA-binding domain and its deletion mutant has effects on gene expression that overlap with Nrg1 based on previously reported microarray studies (*García-Sánchez et al., 2005*). We, therefore, propose that Tup1 may bind to Efg1 rather than to Nrg1 and in doing so inhibit Efg1 function (*Figure 10*). This would explain the overlapping expression profiles of Tup1 and Nrg1 and the differential genetic interactions between Tup1-Efg1 and Nrg1-Tup1. To date, no CHiP analysis of either Tup1 or Nrg1 has been reported during in vitro filamentation. Therefore, it is not possible to definitively identify shared direct targets of Tup1 and Nrg1. The overlapping expression profiles are explained by our proposed model which is also consistent with genetic interaction data we report herein. Accordingly, future detailed biochemical and binding studies of the Tup1-Nrg1-Efg1 axis are likely to provide interesting new insights into the molecular mechanisms of filamentation regulation.

The genetic interaction analysis between the core positive and negative regulators of filamentation uncovered the surprising result that Efg1 is dispensable for the expression of hyphae-associated genes following relief of Nrg1 repression. Each of the Efg1 bound genes that were restored to WT expression by deletion of *NRG1* in the *efg1ΔΔ* mutant also contained an NRE (to our knowledge, no comprehensive analysis of Nrg1 binding has been reported). These data suggest that Efg1 functions, at some level, to mediate the relief of Nrg1. Since Efg1 and Nrg1 bind some of the same targets, a potential mechanism would be for Efg1 to directly displace Nrg1 from the promoters of hypha-associated genes and, thereby, set the stage for TFs such as Rob1, Brg1, and likely others to bind and directly activate gene expression. Other mechanisms are possible and, regardless of that specific mechanism, our findings indicate that during hypha-initiation Efg1 indirectly regulates the expression of hyphae-specific genes. Overall, these genetic interaction studies indicate that much remains to be learned about the molecular mechanism by which hyphal formation is initiated in *C. albicans*.

Based on our genetic screens and genetic interaction studies, we have formulated a genetic circuit for the positive and negative regulation of filament initiation in vivo (*Figure 10*). This circuit incorporates previously established concepts and functions derived from in vitro studies as well as the data

from the current study. As outlined above, the most distinct aspect of this model is that Efg1 plays an indirect role in regulating the canonical set of hyphae-specific genes by mediating the relief of Nrg1 suppression, leaving the direct activation of these genes to other TFs. Although we have largely limited our in vivo transcriptional studies to the core TFs involved in filament initiation, we suspect that at least some of the auxiliary regulators also contribute to the regulation of hyphae-specific genes as we propose for Tec1.

In addition to providing new insights into the initiation of filament formation, we have generated the first large-scale data set of TFs that are required for filament elongation (*Lu et al., 2014*). A number of TF mutants selectively affect the elongation phase of filamentation in vivo and at least five of these do not affect elongation in vitro. Hms1 and Lys14 are two TFs that regulate hyphal elongation in vivo but have few or no other in vitro phenotypes (*Pérez et al., 2013*; *Shapiro et al., 2012*). Perez et al. found that both Hms1 and Lys14 are required for systemic infection, suggesting that reduced elongation may contribute to virulence (*Pérez et al., 2013*). Our in vivo transcriptional profiling of the *HMS1* and *LYS14* mutants suggest that they regulate a distinct set of genes relative to the core TFs involved in filament initiation. Future studies will be required to understand the relationship between reduced filament elongation in vivo and virulence.

At least two distinct mechanisms could contribute to the reduced elongation phenotype for a given TF mutant. First, TFs could specifically, and directly, regulate the expression of genes that are uniquely required for hyphal elongation. A prime candidate for this class of TF is Ume6 since it is expressed only during hyphal formation and is required for this process in vitro and in vivo (*Banerjee et al., 2008*). Second, TFs that have general effects on growth under the conditions of hyphae formation would be expected to reduce the rate or limit elongation. We did not do time course analysis in this work and, therefore, it is possible that some of the mutants with reduced apparent filament length would eventually match WT lengths after additional time. From our set of mutants with reduced length, the *sef1ΔΔ* mutant is potential candidate for this class of TF. Sef1 is required for *C. albicans* to respond to the low iron state established by the host during infection. Deletion of *SEF1* leads to reduced growth under low iron conditions (*Chen et al., 2011*) and, therefore, the reduced cellular iron levels in the *sef1ΔΔ* mutant may reduce filament length simply by limiting the growth of the hyphae. Other mechanisms such as septin regulation, Spitzenkorper assembly, activation, and maintenance of polarized growth factors are all undoubtedly active (*Sudbery, 2011*). Additional studies will be required to dissect the molecular mechanisms by which hyphae elongation-related TFs function during this process.

Ume6 plays an interesting role in that its phenotype is limited to filament elongation but it appears to share targets with hyphae-initiating TFs, particularly with respect to canonical hyphae-associated genes. As such, it seems to function like a transitional TF that bridges the initiation and elongation phases. It is likely that Rob1, Brg1, and other TFs with effects on both initiation and elongation may also bridge the two phases. Indeed, Nrg1 and Brg1 form a negative feedback loop during in vitro filamentation that is thought to maintain hyphal elongation (*Lu et al., 2012*; *Cleary et al., 2012*). Additional work will be required to develop a deeper understanding of the transcriptional regulation of the elongation phase and its key targets, particularly since mutants with selective defects in this process have decreased infectivity and virulence.

It is important to consider specific limitations of our approach. First, our transcriptional analyses are based on a focused set of genes known to be regulated by different environmental condition based on in vitro experiments. Our previous reported analysis of these gene has shown that the expression of these genes in kidney, ear, and in our in vitro conditions is reasonably well correlated. However, we may be missing key targets for in vivo filamentation since no genome-wide analysis of gene expression has been performed in the ear. This also means the proportion of genes regulated by a given TF is the proportion of our set of genes and we cannot make conclusions about the totality of genes regulated by a given TF.

Second, we selected a single in vitro filamentation condition to which we compare our in vivo data. There are a wide range of conditions that induce hyphae formation in *C. albicans*. Our selection was based on two primary considerations: (1) RPMI with 10% serum at 37°C is somewhat host-like and widely used in the field for this reason; (2) the extent of filament formation by our WT strain at 4 hr induction matches the extent of filamentation we observed after 24 hr in vivo; (3) as mentioned above the expression profiles in this condition were reasonably well correlated. However, it is almost certain

that if other in vitro conditions were used as a comparator then a different set of results and conclusions would have resulted.

Third, some of our genetic interaction analysis was based on complex haploinsufficiency using double heterozygous mutants. We used this approach because the homozygous mutants of the core regulators have very low levels of filamentation; therefore, the dynamic range available to assay a double homozygous mutant would be unlikely to provide a statistically interpretable result. The limitation is that the double heterozygotes may not be sensitive enough to reveal subtle interactions; as such, this approach is quite specific but is likely to miss subtle genetic interactions. In the case of the combination of positive and negative regulators, these concerns were not present and, therefore, we used double homozygous mutants.

Finally, it is important to consider the generalizability of these data and conclusions to other filament-inducing conditions and, most importantly, other host niches. As discussed above, it is becoming emphatically clear that the regulatory networks governing *C. albicans* filamentation vary from condition to condition and host niche to host niche. For example, the most widely recognized master transcriptional regulator of filamentation, Efg1, is dispensable for this process when the cells are embedded in agar and incubated at 25°C or under hypoxic conditions. From our own work, we have reported that the *brg1ΔΔ* deletion mutant forms robust filaments in oral tissue but has greatly reduced filamentation in the ear model. Our focus on filamentation within the sub-stromal tissue of the ear epithelium is likely to yield results that have both similarities and differences to filamentation at other sites and under other conditions; however, these limitations are true for any study of *C. albicans* filamentation. We assert that the complexity of this key *C. albicans* virulence trait is worthy of additional study and, therein, likely to yield additional insights.

In summary, our systematic genetic analysis of the transcriptional regulation of *C. albicans* filamentation has both confirmed the importance of key regulators identified in vitro and expanded our understanding of the process to include details and concepts that could not have been inferred from in vitro studies alone. The latter emphasizes the utility of studying virulence-associated traits during infection as well as in vitro.

# Methods

**Key resources table**

| Reagent type (species) or resource | Designation | Source or reference | Identifiers | Additional information |
|---|---|---|---|---|
| Strain, strain background (*Candida albicans*) | *C. albicans* strains generated in this work | See strain table in ***Supplementary file 8*** | | |
| Strain, strain background (*Candida albicans*) | *C. albicans* SN background, homozygous TF deletion mutant collection | Fungal Genetics Stock Center (https://www.fgsc.net/) | Deletion set from Oliver Homann | |
| Strain, strain background (*Mus musculus*, female) | DBA2/N (6–12 weeks) | Envigo | | |
| Biological sample | Fetal bovine serum | Gibco/Thermo Fisher (https://www.thermofisher.com/order/catalog/product/26140079) | Cat. #: 26140-079 | |
| Sequence-based reagent | NanoString Gene probe set | NanoString Co. | Custom Probe Set | See ***Supplementary file 3*** for complete list of genes |
| Sequence-based reagent | RPMI1640 Cell culture medium | Gibco/Thermo Fisher (https://www.thermofisher.com/order/catalog/product/11875093) | Cat. #: 11875-093 | |

*Continued on next page*

*Continued*

| Reagent type (species) or resource | Designation | Source or reference | Identifiers | Additional information |
|---|---|---|---|---|
| Commercial assay or kit | RNA extraction kit | Previously been sold by Lucigen and now can be obtained at https://us.vwr.com/store/product/22399774/masterpuretm-complete-dna-and-rna-purification-kit-biosearch-technologies | Cat. #: 76081-748 | |
| Commercial assay or kit | iScript cDNA synthesis kit (Bio-Rad) | https://www.bio-rad.com/en-us/sku/1708891-iscript-cdna-synthesis-kit-100-x-20-ul-rxns?ID=1708891 | Cat. #: 170-8891 | |
| Commercial assay or kit | iQ SYBR Green Supermix (Bio-Rad) | https://www.bio-rad.com/en-us/sku/1708880-iq-sybr-green-supermix-100-x-50-ul-rxns-2-5-ml-2-x-1-25-ml?ID=1708880 | Cat. #: 170-8892 | |
| Software, Algorithm | GraphPad Prism software | GraphPad Prism (https://graphpad.com) | | Version: 9.5.0 (730) |
| Software, Algorithm | NanoString software | https://nanostring.com/products/analysis-solutions/nsolver-advanced-analysis-software/ | | Version: 4.0 |
| Software, Algorithm | ImageJ /FiJi software | https://imagej.nih.gov/ij/download.html | | Version: 1.8.0_322 (64 bit) |

## Strains, cultivation conditions, and media

The reference and library mutant strains, unless otherwise mentioned, used in this study are derived from the SN background (*Homann et al., 2009*; *Noble and Johnson, 2005*) and were obtained from the Fungal Genetics Stock Center. The *C. albicans* clinical isolate strain and their respective mutants used in this study have been described previously (*Wakade et al., 2021*; *Hirakawa et al., 2015*; *Huang et al., 2019*). All *C. albicans* strains were precultured overnight in yeast peptone dextrose (YPD) medium at 30°C. Standard recipes were used to prepare synthetic drop-out media and YPD (*Homann et al., 2009*). RPMI medium was purchased and supplemented with FBS (10%, vol/vol). For in vitro hyphal induction, *C. albicans* strains were incubated overnight at 30°C in YPD media, harvested, and diluted into RPMI+10% serum at a 1:50 ratio and incubated at 37°C for 4 hr (*Wakade et al., 2021*).

## Mouse experiments

The mouse experiments were approved by the University of Iowa IACUC (protocol number: 0092064/exp. 10/12/2023).

## Strain construction

Strains not associated with the library are summarized in *Supplementary file 8*. *C. albicans* transformations were performed using the standard lithium acetate transformation method (*Noble and Johnson, 2005*). The single and double homozygous mutant strains of *C. albicans* were constructed from an SN152 background using the transient CRISPR/Cas9 method (*Min et al., 2016*). Oligonucleotides and plasmids used to generate the mutant strains in this study are listed in *Supplementary file 9*. Briefly, the *ume6ΔΔ* mutant strain was generated by deleting one copy of *UME6* with *HIS1* cassette which was amplified from pFA-LHL plasmid (*Dueñas-Santero et al., 2019*) with primer pairs UME6.P1 and UME6.2. The second allele of *UME6* was replaced with the *ARG4* marker amplified from pFA-LAL (*Min et al., 2016*) plasmid with primer pairs UME6.P1 and UME6.P2 and by using sgRNA targeting individual alleles of *UME6* gene.

The *brg1ΔΔ* mutant strain was generated by amplifying *ARG4* cassette from pSN69 plasmid (*Noble and Johnson, 2005*) with primer pairs BRG1.P1 and BRG1.P2 and by using sgRNA targeting two alleles of *BRG1* gene. The *rob1ΔΔ* homozygous strain was generated by amplifying *HIS1* cassette from the pSN52 plasmid (*Noble and Johnson, 2005*) with primer pairs ROB1.P1 and ROB1.P2 by using sgRNA targeting two alleles of *ROB1* gene. The resulting *brg1ΔΔ* and *rob1ΔΔ* mutants were further used to generate double homozygous *brg1ΔΔ nrg1ΔΔ* and *rob1ΔΔ nrg1ΔΔ*. To do this, both copies

of *NRG1* knocked out by amplifying *HIS1* or *ARG4* cassette from the plasmid pFA-LHL or pFA-LAL, respectively, with primer pairs NRG1.P1 and NRG1.P2 and using sgRNA targeting two alleles of *NRG1* gene. The *efg1ΔΔ* mutant strain from Homann collection (*Homann et al., 2009*) was used to generate the double homozygous *efg1ΔΔ nrg1ΔΔ* mutant. To do this, pFA-LAL plasmid was used to amplify the *ARG4* cassette using NRG1.P1 and NRG1.P2 primer pairs and using sgRNA targeting two alleles of *NRG1* gene. The resultant transformants were selected on the SD plates lacking either histidine or arginine. The single or double homozygous integration of the deletion cassette was confirmed by standard PCR methods.

Fluorescently labeled strains were generated by using p*ENO1-NEON-NAT1* and p*ENO1-iRFP-NAT1* plasmids as previously described (*Bergeron et al., 2017*; *Seman et al., 2018*) and the resultant transformants were selected on YPD containing 200 µg/ml nourseothricin (Werner Bioagents, Jena, Germany). The reference strain was tagged with green fluorescent protein (NEON) whereas all the TF mutants were tagged with iRFP.

## Inoculation, imaging, and scoring

The inoculation and imaging of mice ear (female DBA/2 mice; 6–12 weeks) were carried out as described previously (*Wakade et al., 2021*; *Wakade et al., 2022b*). Acquired multiple Z stacks (minimum 15) were used to score the yeast vs. filamentous ratio. The cells were considered as a 'yeast' if the cells were round and/or budded cells. Furthermore, yeast cells were required not to project through multiple Z stacks. The cells were considered as a 'filamentous' if the cells contain intact mother and filamentous which was at least twice the length of the mother body. A minimum of 100 cells from multiple fields were scored. Paired Student's t test with Welch's correction (p>0.05) was used to define the statistical significance which was carried out using GraphPad prism software.

## Filament length measurement

Filament length of the in vivo samples were measured as described previously (*Cao et al., 2017*). Briefly, a Z stacks image of the reference or mutant strain were opened in an ImageJ software and the distance between mother neck to the tip of the filament was measured. At least 50 cells per each strain from multiple fields were measured. A statistical significance was determined by Mann-Whitney U test (p>0.05).

## In vitro and in vivo RNA extraction

In vitro and in vivo RNA extraction was carried out as previously described (*Wakade et al., 2022a*; *Xu et al., 2015*). Briefly, for in vitro RNA extraction three independent samples were grown overnight in YPD at 30°C, harvested, and diluted at 1: 50 ratios into the RPMI+10% serum and incubated for 4 hr at 37°C. Cells were collected, centrifuged for 2 min at 11 K rpm at room temperature (RT), and RNA was extracted according to the manufacturer's protocol (MasterPure Yeast RNA Purification Kit, Cat. No. MPY03199). Extraction of RNA from mouse ear was carried out exactly as described previously (*Wakade et al., 2022a*). Briefly, after 24 hr post injection, mouse was euthanized following the protocol approved by the University of Iowa IACUC. The *C. albicans* injected mouse ear was removed and placed into the ice-cold RNA later solution. The ear was then transferred to the mortar and flash-frozen with liquid nitrogen and ground to the fine powder. The resulting powder was collected into 5 ml centrifuge tube and 1 ml of ice-cold Trizol was added. The samples were placed on a rocker at RT for 15 min and then centrifuged at 10 K rpm at 4°C for 10 min. The cleared Trizol was collected without dislodging the pellet into 1.5 ml Eppendorf tube and 200 µl of RNase-free chloroform was added to each sample. The tubes were shaken vigorously for 15 s and kept at RT for 5 min followed by centrifuge at 12 K rpm at 4°C for 15 min. The cleared aqueous layer was then collected to a new 1.5 ml Eppendorf tube and RNA was further extracted following the Qiagen RNeasy kit protocol.

## NanoString analysis

NanoString analysis was carried out as described previously (*Wakade et al., 2022a*). Briefly, in total, 40 ng of in vitro or 1.4 µg of in vivo RNA was added to a NanoString codeset mix and incubated at 65°C for 18 hr. After hybridization reaction, samples were proceeded to nCounter prep station and samples were scanned on an nCounter digital analyzer. nCounter.RCC files for each sample were imported into nSolver software to evaluate the quality control metrics. Using the negative control

probes the background values was defined and used as a background threshold and this value is subtracted from the raw counts. The resulting background subtracted total raw RNA counts were first normalized against the highest total counts from the biological triplicates and then to the WT samples. The statistical significance of changes in gene expression was determined using the Benjmini-Yekutieli procedure with false discovery rate (FDR) <0.1 used as limit of statistical significance. The expression data are summarized in *Supplementary files 3-7*.

### Quantitative reverse transcription-PCR

Precultured ON strains in YPD at 30°C were back-diluted into fresh YPD and collected at the mid-log phase. RNA was isolated using a MasterPure Yeast RNA Purification Kit (Cat. No. MPY03199) and reverse-transcribed using an iScript cDNA synthesis kit (170-8891; Bio-Rad). The qPCR was performed using IQ SyberGreen supermix (170-8892; Bio-Rad) and primers used in this study are listed in *Supplementary file 9*. Briefly, each reaction contained 10 µl of the SYBER Green PCR master mix, 0.10 µM of forward and reverse primer, and 150 ng of cDNA. Data analysis was performed using $2^{-\Delta\Delta CT}$ method and *ACT1* was used as an internal control.

## Acknowledgements

The authors would like to thank Scott Moye-Rowley (Iowa), Scott Filler (UCLA), and Aaron Mitchell (Georgia) for helpful discussions. This work was funded by the National Institute of Allergy and Infectious Diseases grant R01AI133409 (DJK).

## Additional information

### Funding

| Funder | Grant reference number | Author |
|---|---|---|
| National Institute of Allergy and Infectious Diseases | R01AI133409 | Damian J Krysan |

The funders had no role in study design, data collection and interpretation, or the decision to submit the work for publication.

### Author contributions

Rohan S Wakade, Data curation, Formal analysis, Investigation, Methodology, Writing – review and editing; Laura C Ristow, Formal analysis, Investigation; Melanie Wellington, Conceptualization, Formal analysis, Supervision, Methodology, Writing – review and editing; Damian J Krysan, Conceptualization, Formal analysis, Supervision, Funding acquisition, Writing – original draft, Writing – review and editing

### Author ORCIDs

Damian J Krysan http://orcid.org/0000-0001-6330-3365

### Ethics

This study was performed in strict accordance with the recommendations in the Guide for the Care and Use of Laboratory Animals of the National Institutes of Health. All of the animals were handled according to approved institutional animal care and use committee (IACUC) protocols at the University of Iowa as protocol 0092064.

### Decision letter and Author response

Decision letter https://doi.org/10.7554/eLife.85114.sa1
Author response https://doi.org/10.7554/eLife.85114.sa2

# Additional files

## Supplementary files

• Supplementary file 1. Analysis of the extent of filamentation of 155 transcription factor deletion mutants during ear infection. The columns indicate: gene name; percentage filaments formed relative to co-infecting wild type (WT) strain; standard deviation; p value for comparison of the mutant to WT using paired Student's t test.

• Supplementary file 2. Analysis of filament length of 155 transcription factor deletion mutants during ear infection. The columns indicate: gene name; mean length of the wild type (WT); mean length of the mutant; p value for comparison of WT to mutant by Mann-Whitney U test; and relative length of the mutant with its co-infecting WT set to 100%. Bold indicates mutants with statistically different filament lengths (p<0.05).

• Supplementary file 3. NanoString expression profile of efg1ΔΔ, brg1ΔΔ, and rob1ΔΔ mutants in vitro and in vivo. The gene names; raw nCounter counts for wild type (WT) and the mutant strains; normalized counts; average counts for the strains; counts normalized to WT; fold-change for the gene expression in the mutant relative to WT; the p value calculated using Student's t test; and the false discovery rate (FDR) calculated using the Benjamini-Yekutieli procedure are provided. Red indicates a gene downregulated by twofold relative to WT with FDR ≤0.1; green indicates a gene upregulated using the same criteria.

• Supplementary file 4. NanoString expression profile of tec1Δ, efg1Δ, and tec1Δ efg1Δ in vitro and in vivo. The gene names; raw nCounter counts for wild type (WT) and the mutant strains; normalized counts; average counts for the strains; counts normalized to WT; fold-change for the gene expression in the mutant relative to WT; the p value calculated using Student's t test; and the false discovery rate (FDR) calculated using the Benjamini-Yekutieli procedure are provided. Red indicates a gene downregulated by twofold relative to WT with FDR ≤0.1; green indicates a gene upregulated using the same criteria.

• Supplementary file 5. NanoString expression profile of filament elongation regulators hms1ΔΔ, lys14ΔΔ, and ume6ΔΔ in vivo. The gene names; raw nCounter counts for wild type (WT) and the mutant strains; normalized counts; average counts for the strains; counts normalized to WT; fold-change for the gene expression in the mutant relative to WT; the p value calculated using Student's t test; and the false discovery rate (FDR) calculated using the Benjamini-Yekutieli procedure are provided. Red indicates a gene downregulated by twofold relative to WT with FDR ≤0.1; green indicates a gene upregulated using the same criteria.

• Supplementary file 6. NanoString expression profile of Nrg1 double mutants in vitro and in vivo. The gene names; raw nCounter counts for wild type (WT) and the mutant strains; normalized counts; average counts for the strains; counts normalized to WT; fold-change for the gene expression in the mutant relative to WT; the p value calculated using Student's t test; and the false discovery rate (FDR) calculated using the Benjamini-Yekutieli procedure are provided. Red indicates a gene downregulated by twofold relative to WT with FDR ≤0.1; green indicates a gene upregulated using the same criteria.

• Supplementary file 7. Summary of all genes differentially in the study.

• Supplementary file 8. Table of C. albicans strains used in this study.

• Supplementary file 9. Table of oligonucleotides and plasmids used in this study.

• MDAR checklist

## Data availability

All data generated or analyzed during this study are included in the manuscript, supporting files, and source data files for Fig. 1, 2, and 8 are provided. Both the raw and processed gene expression data generated by Nanostring are provided in supplementary files 3,4, 5 and 6. No sequencing data was generated in this study.

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
