## [Editor Report]

Candida morphogenesis is important for virulence. This study provides important new information as to how *C. albicans* regulates the switch from budding to hyphal morphology. Their results identify transcription factors involved in the process of hyphal morphogenesis in the host. The results are convincing and will be interesting to scientists in the fields of medical mycology and cell biology.

---

## [Decision Letter]

**Decision letter after peer review:**

Thank you for submitting your article "Intravital imaging-based genetic screen reveals the transcriptional network governing *Candida albicans* filamentation during mammalian infection Running title: Regulation of *C. albicans* filamentation during infection" for consideration by *eLife*. Your article has been reviewed by 2 peer reviewers, and the evaluation has been overseen by a Reviewing Editor and Arturo Casadevall as the Senior Editor. The reviewers have opted to remain anonymous.

Essential revisions:

As you can see below both reviewers were very favorable about your manuscript. However, they identified several areas where clarifications are needed.

1) In my opinion no new experimental work is needed.

2) Please address the reviewers' points below. I believe their comments/criticisms can be addressed with text changes and additional clarifications in the text.

*Reviewer #1 (Recommendations for the authors):*

The authors should make it more clear what is the interpretation for the mutants with a slower rate of hyphal elongation in vivo (Figure 2). The mutants do not appear to show a significantly slower rate of elongation in vitro, suggesting that they do not have a major metabolic problem. Although it is an interesting phenotype, it is not clear if this is just a metabolic issue, or if this indicates that the rate of hyphal elongation is regulated in vivo.

*Reviewer #2 (Recommendations for the authors):*

1. It would help if the authors could discuss several experimental limitations of this study: (1) proportions of genes controlled by specific TFs may change depending on the Nanostring gene set, (2) in vitro and in vivo targets for TFs can vary depending on conditions and host niches; this is discussed somewhat but it could also alter results and their interpretation, (3) limitations of haploinsufficiency vs. other approaches for determining TF interactions.

2. The model in Figure 10 is hard to reconcile with previous work showing that many Nrg1 target genes are also Tup1 targets and previous studies showing that Nrg1 DNA-binding protein recruits the Tup1 corepressor in yeast. As a global corepressor, Tup1 may also have targets that are not related to Nrg1. These finding should be considered with respect to the model in Figure 10.

3. A rationale for studying selected TFs should be provided (lines 361-366).

4. Alternative mechanisms that control *C. albicans* filament elongation and initiation need to be discussed.

5. Lines 153-154: the observation that certain mutants show increased filamentation in vitro but decreased filamentation in vivo is interesting and additional commentary is warranted.

6. Lines 203-204: UME6 has previously been shown to control hyphal elongation in vivo (Banerjee, et al., 2008) so the observations in Figure S1B are not particularly novel or surprising.

---

## [Author Response]

Reviewer #2 (Recommendations for the authors):1. It would help if the authors could discuss several experimental limitations of this study: (1) proportions of genes controlled by specific TFs may change depending on the Nanostring gene set, (2) in vitro and in vivo targets for TFs can vary depending on conditions and host niches; this is discussed somewhat but it could also alter results and their interpretation, (3) limitations of haploinsufficiency vs. other approaches for determining TF interactions.

Yes, we agree and have added a new section to the discussion.

2. The model in Figure 10 is hard to reconcile with previous work showing that many Nrg1 target genes are also Tup1 targets and previous studies showing that Nrg1 DNA-binding protein recruits the Tup1 corepressor in yeast. As a global corepressor, Tup1 may also have targets that are not related to Nrg1. These finding should be considered with respect to the model in Figure 10.

Microarray experiments have indicated considerable overlap in genes differentially expressed in *tup1*∆∆ and *nrg1*∆∆ mutants. However, this does not mean that they are direct targets of those factors. No CHiP analysis has been done of Tup1 and since it lacks a DNA binding domain, this may be technically difficult. Our model in 10 is not inconsistent with deletions of *TUP1* and *NRG1* having similar effects on expression—indeed, it directly implies that. We now discuss this in more detail and propose a potential molecular explanation for our observations that is consistent with both our new genetic interaction data and the literature data. This is at lines 476-487.

3. A rationale for studying selected TFs should be provided (lines 361-366).

We selected *HMS1* and *LYS14* because they were known to have infectivity defects in mouse models; they did not have an effect on in vitro filamentation initiation or elongation. We have revised this description to improve clarity on lines 293 to 302.

4. Alternative mechanisms that control C. albicans filament elongation and initiation need to be discussed.

Please see response 1 to Reviewer 1 public review.

5. Lines 153-154: the observation that certain mutants show increased filamentation in vitro but decreased filamentation in vivo is interesting and additional commentary is warranted.

We agree that this is interesting. However, almost nothing is known about these TFs in vitro. We respectfully submit that this is better left to dedicated experiments in the future because any comments would be highly speculative.

6. Lines 203-204: UME6 has previously been shown to control hyphal elongation in vivo (Banerjee, et al., 2008) so the observations in Figure S1B are not particularly novel or surprising.

Thank you for this comment. It was not our intention to suggest that this was a novel observation. We have revised the rationale for testing this mutant to as follows*:*

“Ume6 is a TF that is required for hyphal elongation in vitro and appears to do so in vivo based on histology (17). Its deletion mutant is not included in the Homann collection (11). We, therefore, constructed the ume6∆∆ mutant in the SN background to determine if the reduced hyphal elongation phenotype was also observed in our system.”

This section is now at line 169-172.